# Loss of FIC-1-mediated AMPylation activates the UPR^ER and upregulates cytosolic HSP70 chaperones to suppress polyglutamine toxicity

Kate M. Van Pelt[1,2], Matthias C. Truttmann [1,2,3]*

**1** Department of Molecular & Integrative Physiology, University of Michigan, Ann Arbor, United States of America, **2** Program in Cellular & Molecular Biology, University of Michigan, Ann Arbor, United States of America, **3** Geriatrics Center, University of Michigan, Ann Arbor, United States of America

* mtruttma@med.umich.edu

## Abstract

Targeted regulation of cellular proteostasis machinery represents a promising strategy for the attenuation of pathological protein aggregation. Recent work suggests that the unfolded protein response in the endoplasmic reticulum (UPR^ER) directly regulates the aggregation and toxicity of expanded polyglutamine (polyQ) proteins. However, the mechanisms underlying this phenomenon remain poorly understood. In this study, we report that perturbing ER homeostasis in *Caenorhabditis elegans* through the depletion of either BiP ortholog, *hsp-3* or *hsp-4,* causes developmental arrest in worms expressing aggregation-prone polyQ proteins. This phenotype is rescued by the genetic deletion of the conserved UPR^ER regulator, FIC-1. We demonstrate that the beneficial effects of *fic-1* knock-out (KO) extend into adulthood, where the loss of FIC-1-mediated protein AMPylation in polyQ-expressing animals is sufficient to prevent declines in fitness and lifespan. We further show that loss of *hsp-3* and *hsp-4* leads to distinct, but complementary transcriptomic responses to ER stress involving all three UPR^ER stress sensors (IRE-1, PEK-1, and ATF-6). We identify the cytosolic HSP70 family chaperone *F44E5.4*, whose expression is increased in *fic-1-*deficient animals upon ER dysregulation, as a key effector suppressing polyQ toxicity. Over-expression of *F44E5.4*, but not other HSP70 family chaperones, is sufficient to rescue developmental arrest in polyQ-expressing embryos upon *hsp-3* knock-down. We further show that knock-down of *ire-1* or *atf-6* blocks the upregulation of *F44E5.4* in *fic-1-*deficient worms. Taken together, our findings support a model in which the loss of FIC-1-mediated AMPylation engages UPR^ER signaling to upregulate cytosolic chaperone activity in response to polyQ toxicity.

**Data availability statement:** The raw data underlying all findings presented in this study are available as Excel tables located in the Supporting Information. Supplementary Data S1 contains all raw data for Main Text figures, and Supplementary Data S2 contains all raw data for Supplementary Figures. RNA-sequencing data (raw and normalized counts, design matrix) and an R markdown file containing the code used for analysis are publicly available on GitHub (github.com/mtruttma/polyQ-fic1).

**Funding:** KVP was supported by NIH T32 GM007315-43 and by the National Institute of Neurological Disorders and Stroke under award number F31NS127485. MCT was supported by an Alzheimer's Association Young Investigator Award, a Ruth K. Broad Foundation Award, the UM Paul F. Glenn Center for Biology of Aging Research, and by the National Institute of General Medical Sciences under award number 1R35GM142561. The funders had no role in study design, data collection and analysis, decision to publish, or preparation of the manuscript.

**Competing interests:** The authors have declared that no competing interests exist.

## Author summary

The maintenance of a functional proteome is essential for the proper function of a cell – a concept broadly referred to as "proteostasis", or protein homeostasis. In certain diseases, such as Alzheimer's or Huntington's, proteostasis is impaired, resulting in the build-up of misfolded proteins. One strategy to mitigate this is to regulate the key players that maintain proteostasis in the cell, chaperone proteins. In this study, we show that disrupting proteostasis in the endoplasmic reticulum (ER), an organelle heavily involved in protein production, is lethal in developing *C. elegans* nematodes expressing aggregating polyglutamine (polyQ) proteins. This lethality is rescued by the loss of the enzyme FIC-1, which post-translationally regulates the ER-resident chaperones HSP-3 and HSP-4. Using RNA-sequencing, we find that, when either *hsp-3* or *hsp-4* is knocked-down by RNAi, *C. elegans* deficient in *fic-1* activate the unfolded protein response (UPR^ER) to suppress polyQ toxicity. We further show that the UPR^ER-mediated upregulation of the cytoplasmic chaperone *F44E5.4*, is sufficient to rescue polyQ toxicity, while blocking UPR^ER signaling suppresses this rescue. Taken together, our findings reveal a pathway in which FIC-1-mediated regulation of the UPR^ER modulates proteostasis activity in the cytosol to combat misfolded proteins.

## Introduction

The maintenance of cellular proteostasis is a crucial task carried out by a coordinated ensemble of molecular chaperones, stress-response pathways, and protein degradative machinery [1]. The decline of this network is a primary hallmark of aging [2–4] and is responsible for the build-up of misfolded and aggregated proteins characteristic of aging-associated neurodegenerative diseases (NDs) including Alzheimer's disease, Parkinson's disease, and Huntington's disease [5–7]. In these diseases, the gradual disruption of proteostasis facilitates the aggregation of disease-specific proteins from monomers into toxic oligomeric fibrils thought to disrupt cellular homeostasis and drive aging neurons towards degeneration [8–10]. As key members of the proteostasis network, heat-shock proteins (HSPs), particularly Hsp70s and Hsp90s [11], function to intervene in this process, either through direct binding and disaggregation of toxic oligomers [12], sequestration of oligomers into large, insoluble aggregates [13], or by orchestrating the degradation of toxic species through the autophagy-lysosomal or ubiquitin-proteasome pathways [14]. In addition to these direct actions, HSPs also regulate various cellular stress-response pathways [15]. One prominent example is the endoplasmic reticulum (ER)-resident HSP70 chaperone, BiP, which can trigger the ER's unfolded protein response (UPR^ER) and downstream stress-response signaling through activation of UPR^ER sensors IRE1, PERK, and ATF6 [16,17]. Due to their diverse roles as central nodes in the proteostasis network, HSPs have long held interest as druggable targets for the treatment of NDs, particularly through the use of small molecular activators [11,18,19]. However, the

consequences of such a strategy, which include the potential to fuel cancer cells [20], suggest a careful balance must be struck to target discrete HSP populations for the amelioration of ND-associated protein aggregation. As such, proteostasis network components which serve to fine-tune HSP activity represent a promising area for intervention in NDs.

Protein AMPylation is an emerging post-translational modification (PTM) characterized by the addition of an adenosine monophosphate (AMP) moiety from ATP to a serine or threonine side-chain on target proteins. This PTM is performed by single-copy, fic domain-containing enzymes (fic AMPylases) found in most metazoans, including humans *(FICD)*, *Mus musculus (Ficd), Drosophila melanogaster* (Fic), and *Caenorhabditis elegans (fic-1)* [21,22]. Importantly, fic AMPylases are bi-functional, catalyzing both the addition (AMPylation) and removal (deAMPylation) of AMP to substrate proteins [23–25]. Recent literature has coalesced to support a model in which AMPylation predominantly regulates the activity of HSP70 family chaperones in the endoplasmic reticulum (ER) as well as the cytosol [26–31]. While additional targets of fic AMPylases continue to be identified [32–34], the most well-studied target is the ER-resident HSP70 family chaperone Grp78/BiP [29]. BiP is AMPylated in its substrate-free conformation, where the modification prevents ATP hydrolysis and subsequent client binding [35,36]. In this manner, AMPylation is thought to "lock" BiP in an activated, ATP-bound state such that it can immediately engage with and assist client protein re-/folding upon deAMPylation [25]. Similar to the primary target of mammalian AMPylases, the *C. elegans FICD* ortholog, FIC-1, has been shown to preferentially modify HSP70 family chaperones, including the cytosolic HSC70 ortholog, HSP-1, and the two *C. elegans* BiP orthologs, HSP-3 and HSP-4 [26]. These two ER-resident paralogs, HSP-3 and HSP-4, are thought to have arisen through a gene duplication event [37,38], While HSP-3 is expressed constitutively at high levels throughout the worm, HSP-4 exhibits lower basal expression and is highly upregulated in response to stress [39]. Still, previous work has shown that both transcripts are partially regulated by the UPR$^{ER}$ and exhibit some degree of functional compensation, suggesting both overlapping and divergent functions [39,40].

The physiological impacts of changes to cellular fic AMPylase activity are manifold. Over-expression of a constitutively-active fic AMPylase, for example, leads to profound cytotoxicity across a variety of model organisms, including yeast [27], worms [41,42], flies [43], and human cell lines [31,44]. In human patients, mutations in the active site of FICD promote infancy-onset diabetes, severe neurodevelopmental delays [45], and motor neuron disease [46]. Notably, both reported FICD mutations of clinical significance (Arg371Ser, Arg374His) interfere with the enzyme's deAMPylation activity, resulting in abnormally increased levels of AMPylated, inactive BiP when recapitulated *in vitro*. In contrast to excessive fic AMPylation, fic AMPylase deficiency is generally well-tolerated in the absence of stress in several model organisms, including *C. elegans* [26], human cell lines [31], and mice [47–49]. Fic AMPylation is required, however, for regulation of the unfolded protein response in the endoplasmic reticulum (UPR$^{ER}$) in the presence of pharmacological stressors *in vitro* [31] and in the murine exocrine pancreas *in vivo* [48]. Further, Fic AMPylase deficiency reduces neuronal differentiation in human cerebral organoid models [32] and, in dFic knock-out (KO) flies, loss of AMPylation deregulates BiP leading to light-induced blindness and dysregulated visual neurotransmission [50,51]. Recent work has further uncovered a role for fic AMPylation in modulating the aggregation and toxicity of various neurodegeneration-associated protein aggregates (amyloid-β, α-synuclein, polyglutamine (polyQ) repeat proteins) [41]. Taken collectively, these studies highlight a cellular context- and stressor-specific regulatory role for fic AMPylase activity. Despite this, however, a detailed, mechanistic account of how fic AMPylases fine-tunes the proteostasis network in response to stress is lacking.

In this study, we investigate how changes in levels of FIC-1-mediated protein AMPylation directly alter the aggregation and toxicity of proteins containing expanded polyglutamine (polyQ) peptides in *C. elegans*. We show that, when ER homeostasis is impaired by depletion of either BiP ortholog, *hsp-3* or *hsp-4*, worms expressing aggregating polyQs exhibit larval arrest that is rescued by ablation of the AMPylase *fic-1*. We establish that *fic-1* deletion protects against progressive declines in polyQ worm fitness and longevity when ER function is compromised in adulthood. Using bulk RNA-sequencing (RNAseq), we profile the transcriptomic response of polyQ worms lacking *fic-1* to *hsp-3* and *hsp-4* depletion, revealing distinct, cooperative patterns of gene expression. Loss of *hsp-3* yields robust upregulation of oxidative stress-protective

glutathione transferase genes and activation of numerous molecular chaperones and small heat-shock proteins. In contrast, *hsp-4* knock-down elicits a transcriptional program consisting of UPR[ER] and ER-associated degradation (ERAD) genes, as well as changes in N-linked protein glycosylation. Extending these findings, we show in functional assays that signaling through all three UPR[ER] stress sensors (IRE-1, PEK-1, and ATF-6) contributes to the dampening of polyQ toxicity in AMPylation-deficient animals. From our RNAseq analysis, we identify the cytosolic HSP70 family chaperone, *F44E5.4*, as a critical suppressor of polyQ toxicity, and demonstrate that signaling predominantly through IRE-1, as well as ATF-6, is required for the induction of *F44E5.4* in response to ER stress. Whole-body over-expression of *F44E5.4*, but not the HSP70 family chaperones *hsp-1* or *C12C8.1*, phenocopies the effects of *fic-1* deletion in polyQ animals. Mechanistically, we define a model in which, in the absence of FIC-1, UPR[ER] signaling through all three branches responds to ER stress by upregulating *F44E5.4* and other stress-responsive genes to suppress polyQ protein toxicity. Taken as a whole, our findings show that changes in AMPylation levels directly alter the toxicity of neurodegenerative disease-associated proteins and highlight AMPylation as a key regulatory element in the proteostasis network whose function may be exploited to mitigate proteotoxicity in neurodegenerative disease.

## Materials and methods

### *C. elegans* strains and growth conditions

All *C. elegans* strains used in this study were maintained at 20ºC on nematode growth medium (NGM) plates seeded with OP50–1 *E. coli* bacteria as a food source [52]. The names, genotypes, and sources of all strains used in this study are described in S1 Table.

### Plasmid construction

The plasmid (pMT686) for whole-body over-expression was constructed by cloning the ubiquitously-expressed *eef-1A.1* (formerly *eft-3*) promotor into the plasmid pPD117.01 (A gift from Andrew Fire, Addgene plasmid #1587) for expression in *C. elegans*. *F44E5.4*, *C12C8.1*, and *hsp-1* full-length transcripts were amplified from worm cDNA with primers designed to install an N-terminal HA tag. These inserts were then cloned into pMT686 via Gibson Assembly and the resulting plasmids were screened using Sanger sequencing to validate construction. All primer sequences used in this study are listed in S2 Table.

### Generation of transgenic strains

Plasmid constructs for over-expression of *F44E5.4*, *C12C8.1*, and *hsp-1* were injected into wild-type (N2) hermaphrodite worms alongside the co-injection marker, *Pmyo-2::GFP,* to generate strains carrying extrachromosomal arrays. All microinjections were performed by SunyBiotech (Fujian, China). Subsequently, extrachromosomal arrays were integrated by UV irradiation. All strains carrying integrated arrays were back-crossed at least 5x to N2s prior to use in experiments.

### *In vitro* AMPylation reactions

Purification of recombinant HSP-3, HSP-4, and FIC-1(E274G)$_{134-508}$ was performed according to previously described methods [26,44]. For *in vitro* AMPylation of HSP-3 and HSP-4, recombinant FIC-1(E274G)$_{134-508}$ was first incubated in AMPylation reaction buffer [10 mM HEPES (pH 7.5), 7.5 mM MgCl$_2$, 150 mM NaCl, 1 mM DTT, 0.5 mM ATP] and incubated at 20ºC for 30 minutes. Then, recombinant HSP-3 or HSP-4 was added to the reaction, and the mixture was incubated at 20ºC for an additional 60 minutes. Fresh samples were immediately submitted to the Proteomics Resource Facility at the University of Michigan for proteomic analysis using their in-solution digestion protocol. Representative figures depicting the identified AMPylated residues were generated in pyMOL using AlphaFold2 [53]-predicted structures obtained from the AlphaFold Protein Structure Database [54,55].

## AMPylation mapping by liquid chromatography tandem mass spectrometry (LC-MS-MS)

Briefly, cysteines were reduced with 5 mM DTT for 30 minutes at 45°C. Samples were cooled to room temperature and alkylation of cysteines was achieved by incubating with 65 mM 2-Chloroacetamide, under darkness, for 30 min at room temperature. An overnight digestion with approximately 1 µg sequencing grade modified trypsin was carried out at 37°C with constant shaking in a Thermomixer. Digestion was stopped by acidification and peptides were desalted using Sep-Pak C18 cartridges using manufacturer's protocol (Waters). Samples were completely dried using a vacufuge. Resulting peptides were dissolved in 9 µl of 0.1% formic acid/2% acetonitrile solution and 2 µls of the peptide solution were resolved on a nano-capillary reverse phase column (Acclaim PepMap C18, 2 micron, 50 cm, ThermoScientific) using a 0.1% formic acid/2% acetonitrile (Buffer A) and 0.1% formic acid/95% acetonitrile (Buffer B) gradient at 300 nl/min over a period of 180 min (2–25% buffer B in 45 min, 25–40% in 5 min, 40–90% in 5 min followed by holding at 90% buffer B for 5 min and requilibration with Buffer A for 30 min). Eluent was directly introduced into *Q exactive HF* mass spectrometer (Thermo Scientific, San Jose CA) using an EasySpray source. MS1 scans were acquired at 60K resolution (AGC target = 3x10$^6$; max IT = 50 ms). Data-dependent collision induced dissociation MS/MS spectra were acquired using Top speed method (3 seconds) following each MS1 scan (NCE ~ 28%; 15K resolution; AGC target 1x10$^5$; max IT 45 ms).

Proteins were identified by searching the MS/MS data against *E coli BL21 protein database* (4156 entries; uniprot-proteome_UP000002032_Ecoli_BL21.fasta, downloaded on 07/17/2019) appended with protein sequences for HSP-3 (WormBase ID: CE08177), HSP-4 (WormBase ID: CE07244), and FIC-1(E274G) using Proteome Discoverer (v2.4, Thermo Scientific). Search parameters included MS1 mass tolerance of 10 ppm and fragment tolerance of 0.2 Da; two missed cleavages were allowed; carbamidimethylation of cysteine was considered fixed modification and oxidation of methionine, phosphoadenosine (329.053 Da) on histidine, lysine, threonine and tyrosine, deamidation of asparagine and glutamine were considered as potential modifications. False discovery rate (FDR) was determined using Percolator and proteins/peptides with a FDR of ≤1% were retained for further analysis.

## Worm synchronization

Asynchronous worm populations were washed off of NGM plates with M9 buffer [22.1 mM KH$_2$PO$_4$, 42.3 mM Na$_2$HPO$_4$, 85.6 mM NaCl, 1 mM MgSO$_4$] and collected in 15 mL conical tubes. Worms were centrifuged at 215 x *g* in a Sorvall Legend RT centrifuge for 1 minute, M9 was aspirated, and worms were washed once with 5 mL of M9 buffer prior to bleaching. Worms were dissolved in 5 mL of hypochlorite bleaching buffer at 20ºC while rotating on a nutator (Clay Adams) for exactly 6 minutes and bleaching-resistant embryos were subsequently recovered by centrifugation at 215 x *g* for 1 minute. After removing the bleaching solution, embryos were washed twice with 5 mL of M9 buffer and centrifuged at 484 x *g* for 3 minutes before being transferred to fresh NGM plates for downstream experiments. To avoid unwanted selection effects of hypochlorite treatment, synchronized worms were used solely for experimentation and worm stock plates were explicitly spared from bleaching treatments.

## RNA interference (RNAi) feeding

RNAi-mediated knock-down by feeding was performed as described previously [56]. Briefly, on the day of the experiment, fresh NGM-RNAi plates supplemented with 1 mM IPTG (Dot Scientific, #DSI5600–25) and 100 µg/mL carbenicillin (GoldBio, #C-103–25) were seeded with HT115 bacteria carrying a pL4440 plasmid expressing double-stranded RNA (dsRNA) against the gene of interest. The pL4440 empty-vector plasmid or pL4440 encoding dsRNA against *pos-1* were used as controls. The *pos-1* gene encodes a cytoplasmic CCCH zinc-finger protein (POS-1) that is essential for cell fate determination during *C. elegans* early embryogenesis [57,58], but dispensable in adult animals [59]. Animals fed bacteria expressing dsRNA against *pos-1* from hatching are unaffected, but produce non-viable progeny, preventing offspring from interfering with experiments [60]. Embryos or animals were transferred to RNAi plates at indicated time-points and were

maintained at 20ºC for all experiments. To achieve combinatorial knock-downs, equal amounts of both HT115-RNAi *E. coli* strains were seeded onto IPTG plates. RNAi bacterial clones were obtained from either the Vidal [57,58,61] or Ahringer [62] RNAi libraries. Refer to S3 Table for a list of all RNAi clones used in this study and their sources.

### Development assays

For each strain of interest, embryos were harvested by hypochlorite treatment of asynchronous worm populations as described above. Embryos were subsequently transferred to small (35mm) NGM-RNAi plates and seeded with HT115-RNAi *E. coli* bacteria as indicated. The number of embryos on each plate was quantified, and development was scored by visual inspection after 72 hours of incubation at 20ºC. Animals were considered developmentally arrested upon failure to reach the L4 larval stage. Three technical replicates (plates) were scored for each condition and a total of three biological replicates were performed for all experiments.

### ER stress assays

For ER stress resistance assays, tunicamycin (Tocris Bioscience, #35-161-0) or thapsigargin (Sigma-Aldrich, #T9033) dissolved in dimethyl sulfoxide (DMSO, Fisher Scientific, #BP231–100) were added ectopically to small (35mm) NGM plates seeded with OP50–1 *E. coli* 24 hours before use. Final concentrations of 1, 2.5, and 5 µg/mL were utilized for tunicamycin assays, while thapsigargin assay plates contained either 2 or 4.5 µM of the compound. An equivalent amount of DMSO was added to plates and used as a control for each set of experiments. To test for the impact of ER stress, animals were subjected to development assays as described above.

### Thrashing assays

Synchronized worm populations were maintained at 20ºC and assessed in thrashing assays at days 3 and 5 of adulthood. Animals were transferred to a 35mm NGM agar plate containing 1 mL of M9 buffer and allowed to acclimatize for 60 seconds. Then, the number of full body bends performed in 30 seconds was scored by visual inspection. Three independent biological replicates were performed with at least 10 animals scored per experiment.

### Lifespan assays

Day 1 adult animals were transferred to fresh 60mm NGM-RNAi plates seeded with HT115 *E. coli* and moved to fresh plates throughout the first 7 days of adulthood. To avoid unwanted effects of FUDR, RNAi against *pos-1*, a gene whose inactivation prevents egg hatching without affecting adult animals [63], was used at a 50:50 ratio to either empty vector or RNAi against genes of interest. Plates were scored every other day throughout the experiment, and animals were considered dead when repetitive prodding (up to 10x) failed to elicit any detectable body movement. Dead animals were removed immediately from plates upon scoring. All lifespans were conducted at 20ºC.

### Imaging of larval-stage worms and polyQ puncta quantification

L2 larvae were washed off of NGM plates with M9 buffer, collected by centrifugation, and washed 3x with M9 supplemented with 0.01% Trition-X-100. Larvae were then fixed in 4% PFA diluted in M9 for 15 minutes, collected, and washed 3x with M9 supplemented with 0.01% Triton-X-100 to remove fixative. Fixed samples were stored in fresh M9 at 4ºC. For imaging, fixed larvae were pipetted onto glass slides and fluorescent images were obtained using a Leica MZ10-F stereomicroscope equipped with a CCD camera (DFC3000 G, Leica) and pE-3000lite LED illuminator (CoolLED). CellProfiler v3.1.9 [64] was used for the semi-automatic quantification of Q40::YFP puncta number and size. Three independent biological replicates were performed and a minimum of 40 animals per genotype and per condition were assessed.

 

## Adult polyQ worm imaging and quantification

Animals were synchronized by bleaching and grown on NGM-RNAi plates from egg at 20ºC until the indicated time-points. For imaging, worms were first transferred to fresh NGM plates with no bacteria under a brightfield dissecting microscope to avoid sampling bias. Animals were then transferred to 2% agarose pads on glass slides, anesthetized in 10 µL of 25 mM tetramisole hydrochloride, and covered with a glass coverslip for imaging. All images were acquired on a Keyence BZ-X700 All-In-One fluorescence microscope. For analysis, Ilastik v1.4.0.post1 [65] was used to train a machine learning-based segmentation algorithm to identify polyQ puncta from a representative set of training images. The resulting segmentation outputs were further processed in Fiji [66] to collate polyQ puncta number per animal as well as puncta size. Three independent biological replicates were performed and a minimum of 50 animals per genotype and per condition were assessed.

## Immunoblotting

Animals were washed off of NGM plates with M9 buffer, collected by centrifugation, and resuspended in worm lysis buffer (50 mM Tris-HCl pH 8.0, 150 mM NaCl, 5 mM EDTA, 1% NP-40, 0.1% SDS) supplemented with protease and phosphatase inhibitor cocktail (Thermo Scientific). Worms were transferred to 2 mL reinforced microvials (BioSpec) each containing a 5 mm stainless steel bead (Qiagen) and lysed using a Qiagen TissueLyser II (30 Hz, 5 minutes). Samples were centrifuged at 16,100 x $g$ for 15 minutes at 4ºC to pellet worm debris, and the supernatant was collected. Protein concentrations were normalized using the Pierce bicinchoninic acid (BCA) assay kit (Thermo Scientific, #23225), samples were supplemented with SDS sample buffer, and subjected to SDS-PAGE. Proteins were then transferred to activated 0.2 µm polyvinylidene difluoride (PVDF) membrane using the Trans-Blot Turbo RTA Mini Transfer Kit (Bio-Rad, #1704272) and probed with the indicated antibodies. The following antibodies were used in this study: anti-Thr-AMPylation (Biointron, 17G6, mouse mAb); anti-α-tubulin (DSHB Hybridoma Product 12G10, deposited by Frankel, J./ Nelson, E.M., mouse mAb); anti-GFP (Abcam, ab290, rabbit pAb).

## RNA-sequencing and analysis

Animals were synchronized by bleaching and grown at 20ºC on 100mm NGM-RNAi plates seeded with the indicated HT115 *E. coli* strain until the L4 stage of larval development. Animals were washed off of plates with M9 buffer, collected by gravity sedimentation, and subsequently washed 3 times with M9 to ensure complete removal of bacterial food. Worm pellets were collected by centrifugation and stored at -80ºC. Frozen worm pellets were then thawed on ice, resuspended in 300–600 µL of TRI Reagent (Zymo Research, #R2050-1–50) and lysed using a Qiagen TissueLyser II (30 Hz, 5 minutes). RNA isolation was performed using the Direct-Zol RNA Miniprep Plus Kit (Zymo Research, #R2071) according to manufacturer's instructions. Library preparation and sequencing were performed by the Advanced Genomics Core at the University of Michigan. Library preparation was performed using the NEBNext Poly(A) mRNA Magnetic Isolation Module (NEB, #E7490L) and NEBNext Ultra II RNA Library Prep Kit (NEB, #E7770L), and samples were sequenced using an Illumina NovaSeq 6000 S4. Three biological replicates were performed. Reads were mapped to the reference genome WBcel235 using STAR v2.7.8a [67] and assigned count estimates using RSEM v1.3.3 [68]. Differential expression analysis to identify significantly up/down-regulated genes was performed with edgeR (v3.19, Bioconductor) [69]. Venn diagrams used to depict the overlap of enriched genes by genotype and/or RNAi condition were generated using InteractiVenn [70]. Gene ontology (GO) and KEGG pathway analyses were performed using ShinyGO (v0.80) [69].

## Reverse transcription-quantitative PCR

Sample collection and subsequent RNA isolation was performed as described above (see the *RNA-sequencing and analysis* section) and total RNA was reverse transcribed using the High-Capacity cDNA Reverse Transcription Kit

(Applied Biosystems, #4368814). Reactions were prepared with PowerUp SYBR Green Master Mix (Applied Biosystems, #A25742) and analyzed on a StepOnePlus Real-Time PCR System (Applied Biosystems, #4376600). Changes in gene expression were calculated using the 2-ΔΔCT method from three biological replicates per sample, with the housekeeping genes *cdc42* or *pmp-3* being used for normalization [71]. All RT-qPCR primers used in this study are listed in S4 Table.

## Statistical analysis and graphics

All statistical analyses were performed using GraphPad Prism (v10.2.2) software with the exception of the differential expression analysis of RNAseq data (detailed above). The exact statistical tests used are described in the main text figure panels for each dataset. A p-value of $p < 0.05$ was used to determine statistical significance. All graphical illustrations were created in Adobe Illustrator (v29.5.1).

## Results

### The UPR[ER] is required to mitigate polyQ protein toxicity in developing *C. elegan*s.

In previous work, we demonstrated that inducible over-expression of the constitutively-active AMPylase, FIC-1(E274G), in *C. elegans* embryos resulted in complete developmental arrest beyond the L1 larval stage [42]. Changes in endogenous AMPylation levels have also been shown to modulate the aggregation and toxicity of neurodegenerative disease (ND)-associated protein aggregates [41]. We thus wondered if endogenous FIC-1 activity could interfere with proteostasis maintenance in the presence of protein misfolding and aggregation stress during development. To test this, we introduced a *fic-1(n5823)* null allele into a *C. elegans* strain expressing a fluorescently-tagged 40-residue polyglutamine (polyQ) tract in body-wall muscle cells (Q40::YFP) [72]. We then used RNAi to knock down major *hsp70* chaperone genes in the cytosol (*hsp-1)*, and the ER (*hsp-3* or *hsp-4*), to induce protein unfolding stress in wild-type, *fic-1(n5823),* Q40::YFP, and Q40::YFP + *fic-1(n5823)* embryos and assessed their ability to develop into L4 larval animals within 72 hours from hatching (Fig 1A). qPCR confirmed efficient knock-down of the intended target genes by RNAi (S1D - S1F Fig). We found that loss of either *hsp-3* or *hsp-4* did not interfere with larval development in the absence of polyQ aggregation stress. However, in Q40::YFP animals, *hsp-3* or *hsp-4* ablation resulted in complete developmental arrest in the L3 larval stage that was rescued in Q40::YFP animals containing the *fic-1(n5823)* null allele. In contrast, *hsp-1* knock-down was embryonic lethal in all tested genetic backgrounds. Combinatorial knock-down of *fic-1* + *hsp-3* or *fic-1* + *hsp-4* in wild-type or Q40::YFP animals recapitulated this rescue (S1C Fig). Given the limited effect of RNAi in *C. elegans* neurons due to a lack of the dsRNA transporter, SID-1 [73], we posit that the rescue observed here is predominantly non-neuronal.

Based on these results, we next tested whether endogenous HSP-3 and/or HSP-4 AMPylation occurs and is detectable during early larval development. To this end, we prepared lysates from L2 larvae of wild-type, *fic-1(n5823)*, Q40::YFP, and Q40::YFP + *fic-1(n5823)* animals and assessed protein AMPylation by immunoblotting using an antibody specific for AMPylated threonine (Thr-AMP) residues [74] (Figs 1B–1C, S1A–S1B). Under control RNAi (*pos-1*) conditions, we detected robust AMPylation signatures at a molecular weight corresponding to that of HSP-3 and HSP-4. This signal partially decreased upon knock-down of *hsp-3* or *hsp-4*, consistent with AMPylation of both chaperones. To corroborate this approach, we additionally performed *in vitro* AMPylation reactions with recombinant FIC-1(E274G) and HSP-3 or HSP-4 and submitted these samples for analysis by mass spectrometry. This revealed one AMPylated residue on HSP-3 (T194), and three AMPylated residues on HSP-4 (T368, T455, T576) (S3A–S3B Fig). As expected, *fic-1(n5823)* null samples were devoid of detectable Thr-AMP signal (Fig 1B–1C). Interestingly, AMPylation levels were unchanged in the presence of Q40::YFP compared to wild-type samples. This suggests that baseline levels of endogenous AMPylation are sufficient to interfere with Q40::YFP larval development when ER homeostasis is perturbed.

Since polyglutamine expansion diseases exhibit threshold-dependent toxicity around 35–40 residues [75,76], we hypothesized that HSP-3/HSP-4 AMPylation may only be detrimental in the context of longer, aggregating polyQ tracts.

PLOS Genetics

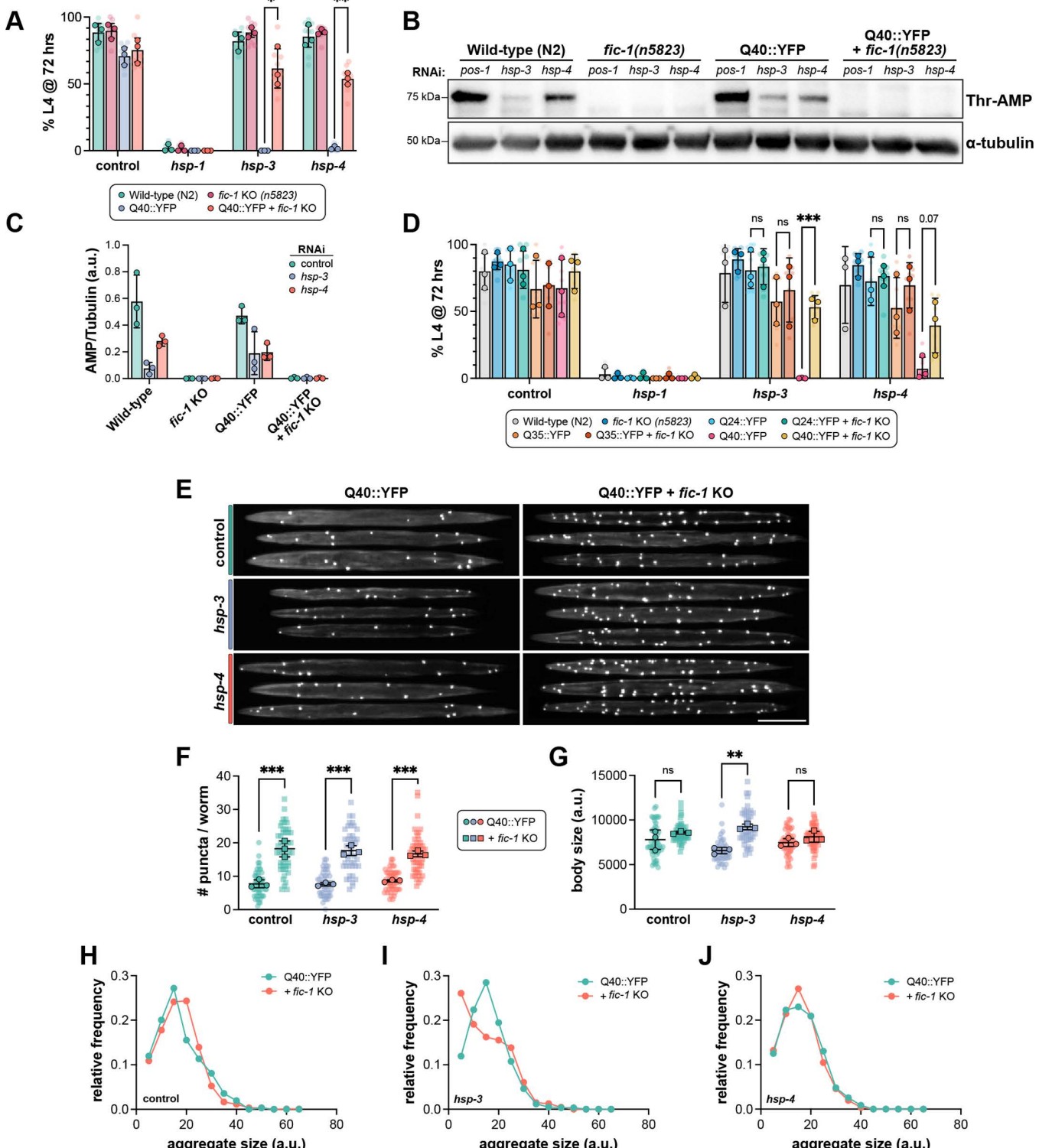

**Fig 1. The UPR^ER is required to mitigate polyQ protein toxicity in developing *C. elegans*.** (A) Development assay of the indicated strains (legend) depicting the percentage of embryos surviving to at least the L4 stage of larval development at 72 hours when fed RNAi against the indicated genes (X-axis) from hatching. (B) Western blot of lysates from L2 larvae fed the indicated RNAis from hatching and probed for Thr-AMP signal (top) and

α-tubulin (bottom) as a loading control. (C) Quantification of Thr-AMP signal intensity normalized to α-tubulin. Each data point represents one lane from one biological replicate. (D) Development assay of the indicated strains depicting the percentage of embryos surviving to at least the L4 stage of larval development at 72 hours when fed RNAi against the indicated genes (X-axis) from hatching. (E) Representative images of L2 larvae following treatment with the indicated RNAis from hatching. Scale bar = 100μm. (F) Quantification of the number of polyQ puncta per animal following RNAi treatment (X-axis). (G) Quantification of L2 larval body size following RNAi treatment. (H-J). Frequency distribution profiles of polyQ puncta sizes in L2 larvae on control (H), *hsp-3* (I), or *hsp-4* (J) RNAi. Bin size = 20 a.u. In (A, D), translucent data points depict technical replicates, while opaque data points reflect the average for each biological replicate performed (n = 3). In (F-G), each translucent data point represents one worm, and opaque data points reflect the average for each biological replicate (n = 3). Error bars for all plots represent SD. Two-way (A, D) or one-way (F-G) ANOVAs with Tukey's post-hoc multiple comparisons tests were performed to determine statistical significance. ***$p < 0.001$; **$p < 0.01$; *$p < 0.05$; ns = not significant.

Using two additional *C. elegans* strains expressing shorter polyQ tracts, Q24::YFP and Q35::YFP, which do not form puncta during larval development [72], we repeated this assay (Fig 1D) in the presence or absence of *fic-1(n5823)*. In line with our hypothesis, *fic-1* deletion had no effect on the development of Q24::YFP or Q35::YFP animals upon *hsp-3* or *hsp-4* knock-down. Instead, the protective effects of *fic-1* loss were specific to Q40::YFP animals, consistent with a polyQ tract length-dependent phenomenon. Expanding upon this, we also tested the impact of two small molecules known to induce ER stress, tunicamycin (Tm), and thapsigargin (Tg) on worm development (S3C–S3D Fig). In the absence of polyQs, *fic-1(n5823)* null animals were less sensitive to Tm treatment than their wild-type counterparts. In the presence of polyQs, however, *fic-1* KO animals exhibited a trend towards increased Tm sensitivity (S3C Fig), suggesting that the benefits of decreased AMPylation decline when multiple proteotoxic insults accumulate.

Having established polyQ toxicity as a driver of embryonic ER stress, we next sought to examine polyQ aggregation dynamics in L2 larvae prior to the onset of developmental arrest. In line with our previous reports in adult worms, *fic-1* KO larvae exhibited increased numbers of Q40::YFP puncta (Fig 1E–1F), an observation that occurred independently of *hsp-3* or *hsp-4* depletion. Interestingly, *hsp-3*, but not *hsp-4* knock-down led to reduced body size in the absence of *fic-1* KO (Fig 1G). While there were no significant differences in the distribution of Q40::YFP puncta sizes upon *hsp-4* knock-down (Fig 1H and 1J), *hsp-3* depletion was associated with a decrease in the relative proportion of smaller puncta in *fic-1* KO larvae (Fig 1I). Of note, we did not detect any significant changes in Q40::YFP expression due to RNAi treatment conditions or the presence of the *fic-1(n5823)* null allele (S2A–S2D Fig). Taken together, these results indicate that the loss of FIC-1-mediated AMPylation impacts polyQ aggregation dynamics and bolsters ER proteostasis during a critical developmental window.

## Loss of FIC-1-mediated AMPylation protects against declines in polyQ worm fitness and lifespan in adulthood

With the knowledge that loss of AMPylation confers protection during worm development in the presence of aggregating polyQs, we next assessed if FIC-1 activity is dispensable later in life in adult animals. To avoid lethality in polyQ worms plated on *hsp-3*/*hsp-4* RNAi from hatching, we first synchronized embryos from all strains on control RNAi and transferred animals to experimental RNAi conditions on day 1 of adulthood (Fig 2A). As a general read-out for overall worm fitness and motility [77,78], we performed thrashing assays in day 3 (Fig 2B) and day 5 (Fig 2C) adult animals. Under control conditions, *fic-1* KO had no impact on Q40::YFP worm motility at either time-point tested. Interestingly, however, UPRER dysregulation through the depletion of either *hsp-3* or *hsp-4* in adulthood led to a progressive decline in thrashing rates of Q40::YFP animals at day 3 and day 5, while Q40::YFP; *fic-1(n5823)* worms were protected from this insult. As a secondary measure, we next performed longevity experiments under the same conditions (Figs 2D - 2F, S4A–S4C). Similar to the results observed in the thrashing assay, *fic-1* KO did not impact Q40::YFP lifespan under control conditions (Figs 2D, S4A). Upon *hsp-3* depletion, however, the lifespan of Q40::YFP animals diminished, while Q40::YFP; *fic-1(n5823)* worms again remained unaffected (Figs 2E; S4B). Knock-down of *hsp-4* decreased lifespan of all strains tested, but here, too, *fic-1* KO led to a small but significant increase in lifespan specific to the presence of polyQs (Figs 2F; S4C). Quantifying polyQ puncta at day 3 (Fig 2G–2H) and day 5 (Fig 2I–2J) of adulthood, *fic-1* KO increased the number of

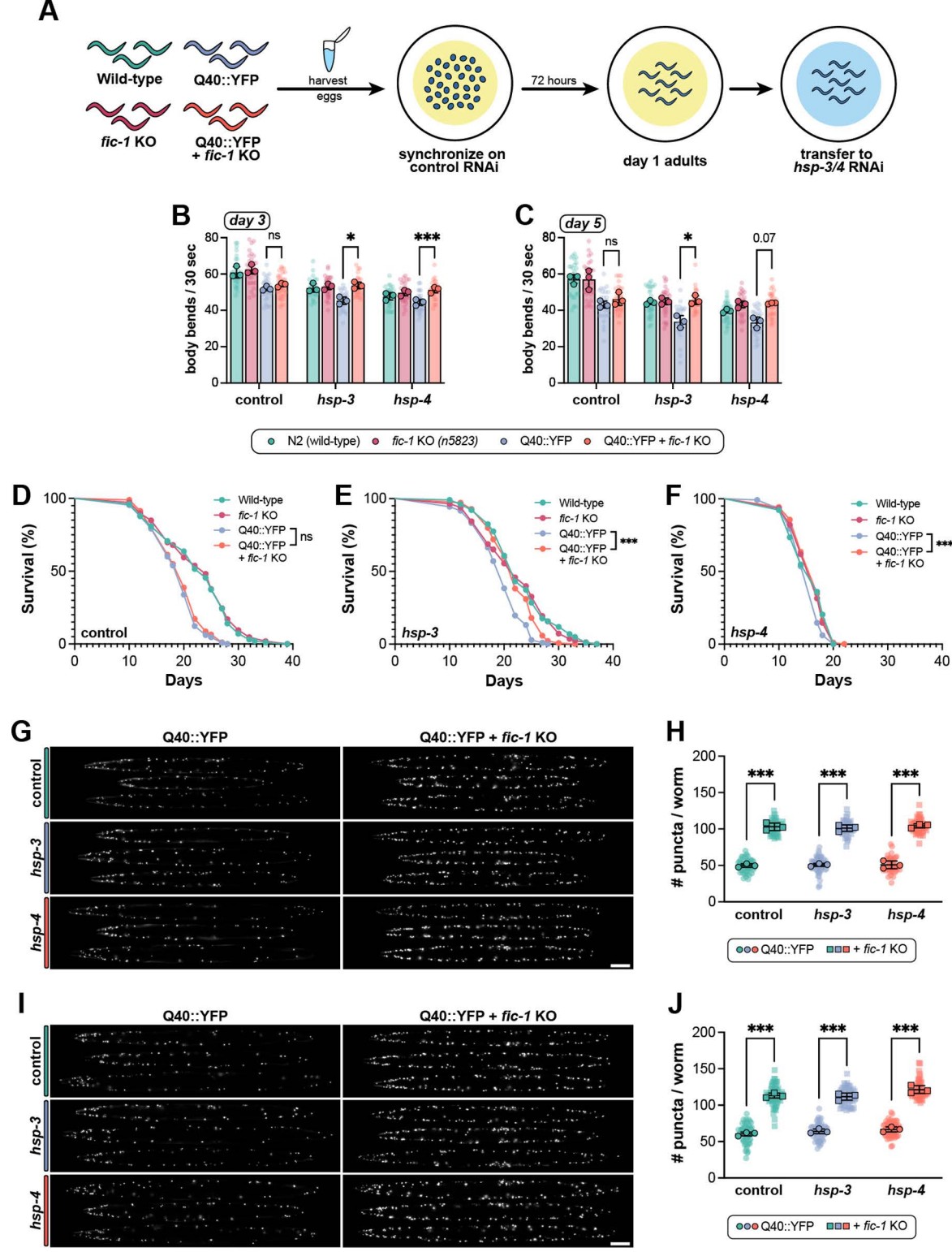

**Fig 2. Loss of FIC-1-mediated AMPylation protects against declines in polyQ worm fitness and lifespan in adulthood.** (A) Schematic outlining RNAi treatment paradigm in adult animals. Embryos were synchronized on control RNAi and transferred at day 1 of adulthood to prevent embryonic

lethality in polyQ worms observed upon *hsp-3/4* loss. (B-C) Quantification of thrashing rates for day 3 (B) or day 5 (C) adult animals of the indicated genotypes (legend) upon treatment with the indicated RNAis (X-axis) beginning at day 1 of adulthood. (D-F) Lifespan experiments of wild-type and polyQ worms in the presence or absence of the *fic-1(n5823)* null allele when fed control (D), *hsp-3* (E), or *hsp-4* (F) RNAi in adulthood. (G-H) Representative images (G) and quantification (H) of polyQ puncta in day 3 adult animals when treated with the indicated RNAis in adulthood. (I-J) Representative images (I) and quantification (H) of polyQ puncta in day 5 adult animals when treated with the indicated RNAis in adulthood. For both (G) and (I), scale bar = 100μm. In (B-C) and (H-J), each translucent data point represents one individual worm, and opaque data points depict the average of each biological replicate (n = 3). Error bars for all plots represent SD. Two-way (B-C) or one-way (H-J) ANOVAs with Tukey's post-hoc multiple comparisons tests were performed to determine statistical significance. For lifespan studies (D-F), a Mantel-Cox test was used. ***p < 0.001; *p < 0.05; ns = not significant.

puncta independent of RNAi condition (Fig 1E–1F). In adult animals, however, Q40::YFP; *fic-1(n5823)* worms exhibited an increased relative frequency of large polyQ puncta, possibly indicating a shift towards sequestration of toxic polyQ species later in adulthood (S4D -S4F and S4G–S4I Fig). Taken as a whole, our data show that, in the context of protein aggregation stress in the ER, the loss of FIC-1-mediated AMPylation has a beneficial influence on worm fitness beyond larval development, imparting cytoprotective effects well into adulthood.

### *fic-1* KO initiates a transcriptional response to counteract protein misfolding stress

To gain further insight into the gene(s) and pathway(s) activated in the absence of FIC-1-mediated AMPylation, we performed bulk RNA sequencing analysis of wild-type, *fic-1(n5823)*, and Q40::YFP ± *fic-1(n5823)* animals under *hsp-3* and *hsp-4* knock-down conditions. Each strain was grown on control, *hsp-3*, or *hsp-4* RNAi from embryo to the L4 stage of larval development with the exception of Q40::YFP, which was grown solely on control RNAi due to the lethal effects of *hsp-3* or *hsp-4* knock-down in this strain (Fig 3A). A principal component analysis of the resulting transcriptomes showed separation of the samples based on both the strains' genetic backgrounds and RNAi treatment conditions (Fig 3B). We first validated the efficacy and specificity of *hsp-3* and *hsp-4* RNAi, both of which robustly suppressed expression of their intended gene targets (S5C–S5E Fig). While differential expression analysis revealed no significant effect of a given RNAi (e.g., against *hsp-3*) on the other ortholog (e.g., *hsp-4*) (S5A–S5B Fig), we did find that knock-down of *hsp-3* increased *hsp-4* transcripts when assessed by qPCR, as has been shown previously [39] (S5C and S5E Fig).

Knock-down of *hsp-3* and *hsp-4* elicited robust changes in gene expression relative to control conditions, with 415 genes significantly upregulated and 363 genes downregulated [cut-offs: p < 0.05; $\log_2$FC > 1.5 or> -1.5, respectively] upon *hsp-3* depletion and 695 genes up- and 784 genes down-regulated under *hsp-4* knock-down across all genotypes. Gene ontology (GO) analysis of commonly upregulated genes revealed an activation of protein folding-related genes upon *hsp-3* knock-down, while *hsp-4* knock-down resulted in the shared upregulation of genes involved in the response to ER stress (S5F–S5G Fig). We then isolated the transcriptomic response specific to Q40::YFP; *fic-1* KO animals for further analysis.

Upon *hsp-3* knock-down, we identified 269 upregulated and 290 downregulated genes unique to Q40::YFP; *fic-1* KO worms (Figs 3C, S6A). Similarly, *hsp-4* knock-down elicited the upregulation of 371 genes and the downregulation of 543 genes (Figs 3F, S6D). GO enrichment analysis for over-represented biological process terms revealed a significant enrichment of protein folding and stress response-related genes upon *hsp-3* loss in Q40::YFP; *fic-1* KO animals, notably including a specific signature related to glutathione metabolism (Fig 3D and 3I). *hsp-4* knock-down resulted in a similar enrichment of stress-response and protein folding genes accompanied by genes specific to the endoplasmic reticulum unfolded protein response (UPR^ER) and ER-associated degradation (ERAD) pathways (Fig 3G). Further examination of GO molecular function terms and enriched KEGG pathways in Q40::YFP; *fic-1* KO animals on *hsp-3* RNAi revealed the upregulation of processes related to unfolded protein binding and ER protein processing, identifying the ER stress response as a shared hallmark of polyQ protein toxicity in the absence of either *hsp-3* or *hsp-4* (Fig 3E and 3H). Selecting genes for further testing, we identified a suite of significantly upregulated molecular chaperones upon *hsp-3* loss, including *hsp-90*, the HSP-90 ATPase activator *ahsa-1*, HSP70 family chaperone members *F44E5.4* and *F44E5.5*, the non-canonical small heat-shock protein (sHSP) *hsp-17*, and the sHSP *hsp-16.49* (Figs 3I, S7A). The activation of numerous

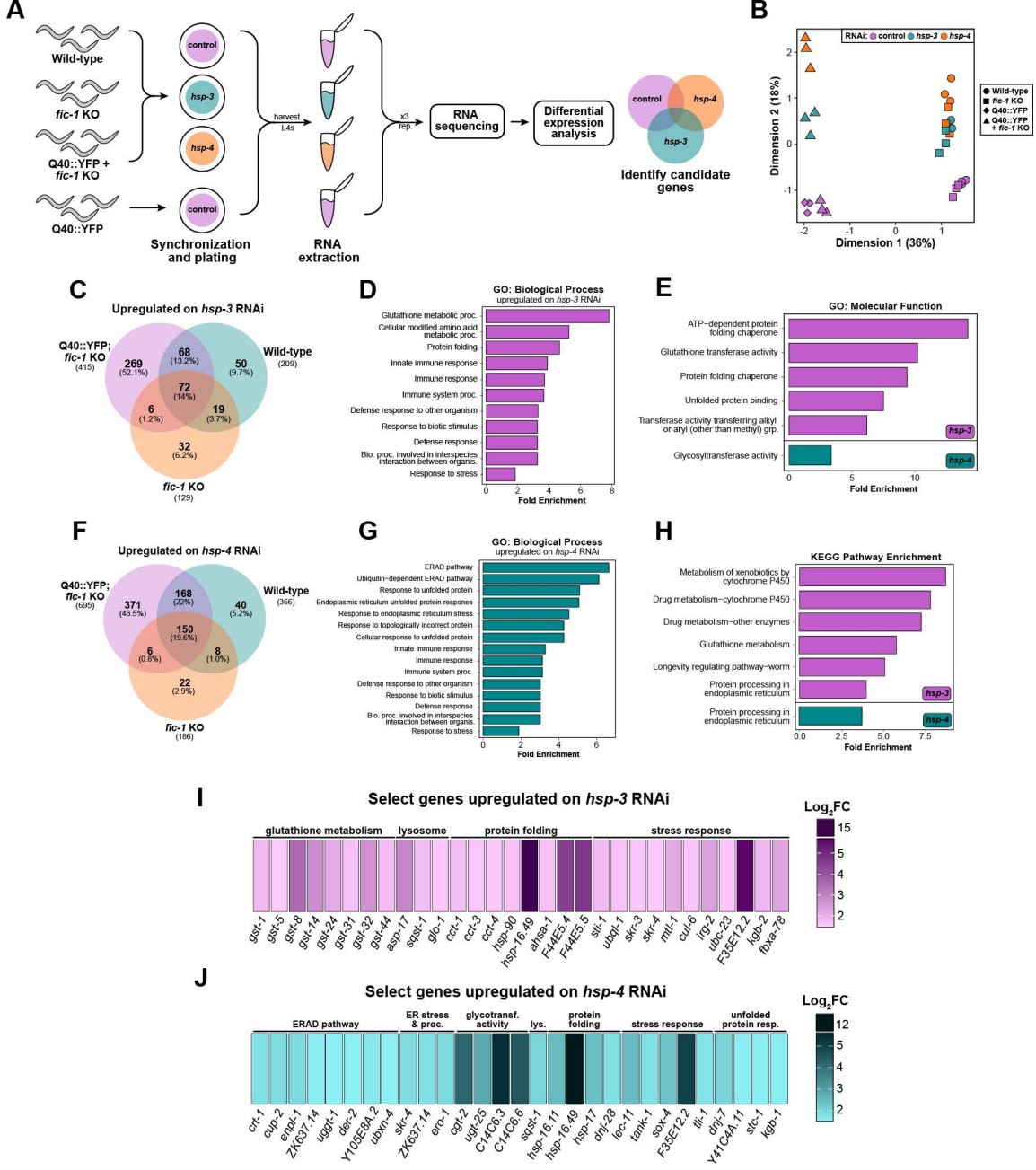

**Fig 3. *fic-1* KO initiates a transcriptional response to counteract protein misfolding stress.** (A) Graphical depiction of experimental design used for RNA sequencing studies. (B) Principle component analysis (PCA) plot showing the separation of samples based on genotype and RNAi treatment conditions. Each data point represents one biological replicate from the indicated group. (C) Venn diagram of genes upregulated upon *hsp-3* knock-down across genotypes, with 269 genes specific to Q40::YFP; *fic-1(n5823)* animals. (D) Over-represented gene ontology (GO) biological process terms under *hsp-3* knock-down conditions ordered by fold enrichment. (E) Over-represented GO molecular function terms under *hsp-3* (magenta; top) and *hsp-4* (teal; bottom) knock-down conditions. (F) Venn diagram of genes upregulated upon *hsp-4* knock-down across genotypes, with 371 genes specific to Q40::YFP; *fic-1(n5823)* animals. (G) Over-represented GO biological process terms under *hsp-4* knock-down conditions ordered by fold enrichment. (H) Enriched KEGG pathways upon *hsp-3* (magenta; top) or *hsp-4* (teal; bottom) knock-down. (I) Curated list of metabolism, lysosomal, protein folding, and stress response-related genes enriched in Q40::YFP; *fic-1(n5823)* animals upon *hsp-3* knock-down. (J) Curated list of ER-associated degradation (ERAD), ER stress and protein processing, glycotransferase, lysosomal, protein folding, stress response, and UPR$^{ER}$-related genes enriched in Q40::YFP; *fic-1(n5823)* animals upon *hsp-4* knock-down. For (I-J), gene heat-maps are colored according to log$_2$FC value (cut-off: log$_2$FC > 1.5; p < 0.05) with darker colors corresponding to larger FC values.

glutathione S-transferases (*gst-1, -5, -8, -14, -24, -31, -32, -44*) was unique to *hsp-3* loss and suggests a possible role for oxidative stress in polyQ toxicity, as reported previously in *C. elegans* [79]. Similar to *hsp-3* loss, *hsp-4* knock-down also resulted in the upregulation of *hsp-17* and hsp-*16.49*, but further involved activation of the sHSP *hsp-16.11*, DnaJ/HSP40 family proteins *dnj-28* and *dnj-7,* HSP70 chaperone *stc-1*, and canonical ERAD genes (*crt-1, cup-2, der-2, enpl-1*) (Figs 3J, S7B). These gene programs activated in response to *hsp-3* or *hsp-4* knock-down were highly specific to Q40::YFP; *fic-1* KO animals, as analysis of upregulated genes unique to wild-type or *fic-1* KO animals under either RNAi condition failed to uncover distinct clusters of related genes or further insights via GO analysis (S8A–S8D Fig).

In a limited screen, we next tested whether RNAi against a subset of the genes upregulated in Q40::YFP; *fic-1* KO animals, either alone or in combination with *hsp-3* or *hsp-4* RNAi, could result in developmental arrest. While no single gene's knock-down fully suppressed larval development, significant impacts were observed for *cct-1* and *cct-4* in combination with *hsp-3* RNAi, and *lec-11* and *enpl-1* in combination with *hsp-4* RNAi (S7C - S7D Fig). Interestingly, genes found to be significantly downregulated upon *hsp-3* or *hsp-4* knock-down were overwhelmingly associated with eggshell formation, oogenesis, and lysosomal function (S6B–S6C and S6E–S6F Fig). In a second limited screen, we tested RNAi against the top downregulated genes in Q40::YFP; *fic-1* KO worms upon *hsp-3* or *hsp-4* loss to assess if their knock-down in Q40::YFP animals could reverse developmental arrest. Strikingly, RNAi against any of the genes tested (*col-135, ilys-5, lys-10, abu-2*) in combination with *hsp-3* RNAi failed to promote survival (S6G Fig). Results were similar for RNAi against genes tested in combination with *hsp-4* RNAi (*cpr-8, vit-5, asp-3, clec-53, asah-1*), suggesting that downregulation of these genes in Q40::YFP; *fic-1* KO animals likely is not a driving survival factor in the face of perturbed ER homeostasis (S6H Fig). Instead, the notable suppression of genes related to reproduction may serve as a complementary protective mechanism or "trade-off" during the final stages of development to cope with protein misfolding stress [80,81].

Taken as a whole, these findings provide a holistic characterization of how the transcriptome of AMPylation-deficient *C. elegans* expressing aggregation-prone polyQs responds to perturbed ER homeostasis. Our data reveal a broad stress-responsive paradigm, highlighting roles for molecular chaperones, ERAD, UPR$^{ER}$, and metabolic genes in mitigating polyQ toxicity in the absence of *fic-1*.

## Loss of *fic-1* activates UPR$^{ER}$ signaling to combat misfolded proteins during development

Following up on the resuts from our RNA sequencing analysis, we first tested for involvement of each UPR$^{ER}$ branch in polyQ toxicity during worm development. In *C. elegans*, the UPR$^{ER}$ consists of three stress sensors, IRE-1, PEK-1, and ATF-6 which are activated in response to protein folding stress and function to initiate downstream upregulation of stress-responsive genes. Given our data implicating UPR$^{ER}$ signaling in the response to polyQ protein toxicity in the absence of *fic-1* (Fig 3E and 3G), we first assessed each branch of the UPR$^{ER}$ using RNAi-mediated depletion in development assays (S9A - S9E Fig). Upon loss of *hsp-3* or *hsp-4*, we observed the expected developmental arrest phenotype in Q40::YFP animals that was rescued by *fic-1* KO. RNAi against *ire-1*, or its downstream mediator, *xbp-1*, had no significant effect on development in wild-type or *fic-1(n5823)* animals in the absence of polyQs (Fig 4A), in line with previous reports [82]. In the presence of Q40::YFP, *ire-1* and *xbp-1* knock-down significantly impeded worm development, with a modest increase in *fic-1* KO animals (Fig 4A), suggesting that the beneficial effects of *fic-1* loss are not solely mediated through this arm of the UPR$^{ER}$. Next, we tested the two other UPR$^{ER}$ branches, PEK-1 and ATF-6. Interestingly, RNAi-mediated knock-down of either *pek-1* or *atf-6* not only impaired development of Q40::YFP-expressing animals, it also abrogated the rescue effects imparted by *fic-1* KO (Fig 4B). Combinatorial knock-down of *pek-1* and *atf-6* did not further reduce worm survival (S9F Fig) suggesting that signaling from either branch is required and sufficient to drive protection from the loss of FIC-1-mediated AMPylation. Given the significant enrichment of UPR$^{ER}$-related genes amongst our RNAseq hits for both *hsp-3* and *hsp-4* knock-down conditions (Fig 3I–3J), we next examined roles for PEK-1's downstream effector proteins, the transcription factor ATF-4 and the translation initiation factor eIF2. Using a combinatorial RNAi approach, we observed that knock-down of either *atf-4* or *eif-2A* blunted the beneficial role of *fic-1* KO during polyQ worm development (Fig 4D), though loss of *eif-2A* was also observed to increase developmental arrest in a non polyQ-dependent manner. Further,

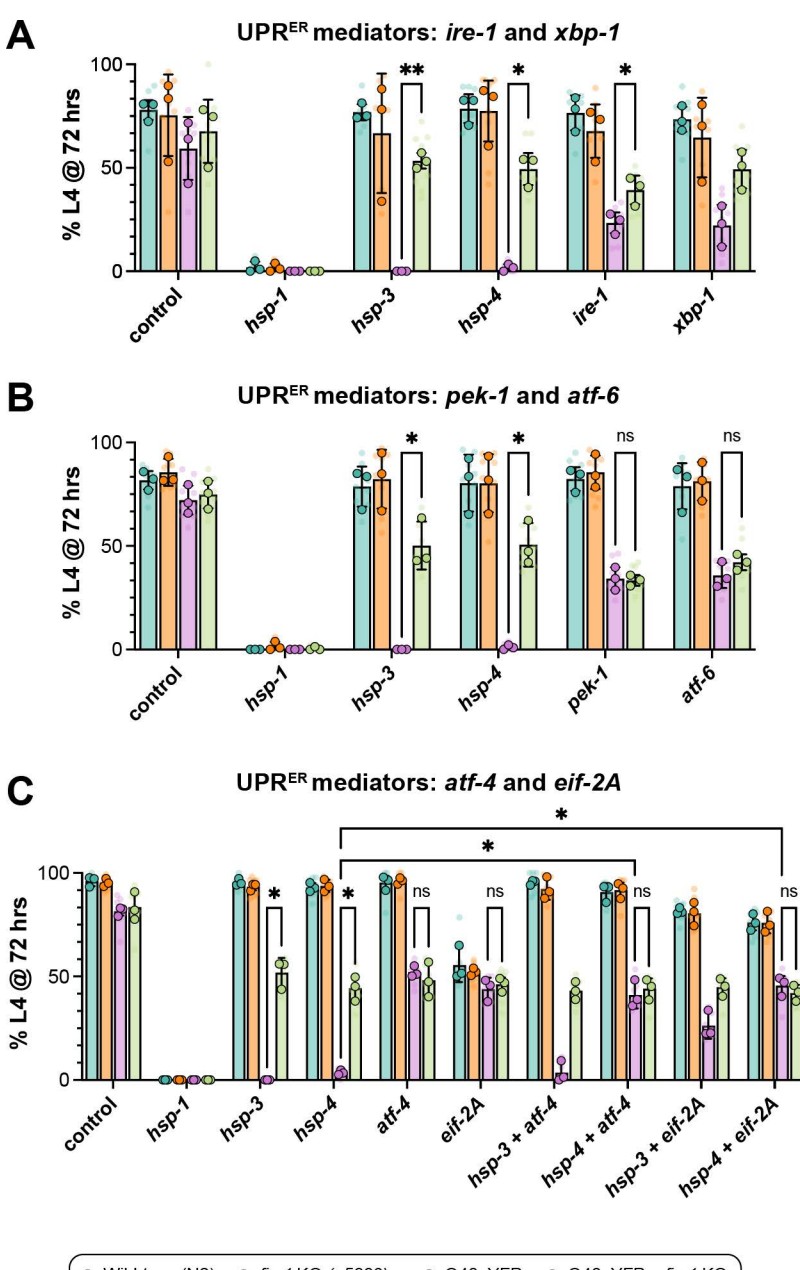

**Fig 4. Loss of *fic-1* activates UPR^ER signaling to combat misfolded proteins during development.** (A) Development assay of the indicated strains assessing for the impact of the IRE-1 branch of the UPR^ER on development. (B) Development assay testing involvement of the PEK-1 and ATF-6 UPR^ER branches. (C) Development assay of downstream *pek-1* mediators, ATF-4 and EIF-2A and their role in polyQ worm development. (A-C) All graphs depict the percentage of animals reaching the L4 stage of larval development at 72 hours. Translucent data points reflect technical replicates (plates), while opaque data points represent biological replicates (n = 3). Error bars represent SD. For all graphs, a two-way ANOVA with Tukey's post-hoc multiple comparisons tests were performed to determine statistical significance. **p < 0.01; *p < 0.05; ns = not significant.

combinatorial knock-down of *hsp-3* and either *atf-4* or *eif-2A* was sufficient to restore *fic-1(n5823)* rescue in Q40::YFP animals, but failed to do so under *hsp-4* knock-down conditions. Taken collectively, these data highlight UPR^ER branch-specific roles for coping with protein folding toxicity during worm development and identify contributions from *ire-1*, *pek-1*, and *atf-6* as key components of this response. Additionally, the *hsp-3*-specific rescue upon concurrent loss with either *atf-4* or *eif-2A* suggests a chaperone-specific effect of downstream *pek-1* signaling, perhaps owed to *hsp-4*'s reported role as a highly stress-responsive ER chaperone.

**UPR^ER signaling upregulates expression of the cytosolic HSP70 family chaperone, *F44E5.4*, to suppress polyQ toxicity**

Having observed a significant upregulation of numerous, predominantly cytosolic, molecular chaperones upon *hsp-3* knock-down, we next asked if enhanced cytosolic chaperone activity is sufficient to phenocopy the rescue obtained by the loss of *fic-1*. To this end, we generated transgenic *C. elegans* strains expressing three key HSP70 family chaperones – *F44E5.4*, *hsp-1*, and *C12C8.1* – under the ubiquitous *eef1A.1* (formerly *eft-3*) promoter for whole-body over-expression (Fig 5A). As transcripts of *F44E5.4* were significantly upregulated in Q40::YFP; *fic-1* KO animals in response to *hsp-3* knock-down (Fig 3I), we first asked whether *F44E5.4* over-expression could promote the survival of Q40::YFP animals placed on *hsp-3* or *hsp-4* RNAi from hatching. Indeed, in development assays, *F44E5.4* over-expression reversed the developmental arrest of Q40::YFP animals grown on either *hsp-3* or *hsp-4* RNAi to the same extent as *fic-1* deletion (Fig 5B). However, neither the over-expression of *hsp-1*, nor *C12C8.1*, imparted the same rescue effect, indicating a unique role for *F44E5.4* in suppressing polyQ toxicity (S10A – S10B Fig). To further probe whether *F44E5.4* upregulation occurs as a direct consequence of *fic-1* loss, we performed development assays knocking-down *F44E5.4* alongside *hsp-3* or *hsp-4* in Q40::YFP; *fic-1* KO animals. While *F44E5.4* RNAi alone did not impact worm survival, combinatorial knock-down of *hsp-3* + *F44E5.4* or *hsp-4* + *F44E5.4* significantly impeded worm development, indicating that *F44E5.4* upregulation contributes to, but is not the sole effector mitigating polyQ toxicity in the absence of *fic-1* (Fig 5C). Assessment of *F44E5.4* knock-down efficacy via qPCR revealed incomplete suppression of *F44E5.4* transcripts (S10E Fig), providing a potential explanation for the observed partial rescue. Conversely, we also investigated the impact of *fic-1* knock-down in Q40::YFP; *F44E5.4* over-expressing (OE) worm development. In this case, *fic-1* RNAi alone did not impact worm development, and combinatorial knock-down of *hsp-3* + *fic-1* or *hsp-3* + *fic-1* failed to elicit any additive effect on worm development (Fig 5D). Together, these findings suggest that enhanced *F44E5.4* expression occurs as a direct consequence of *fic-1* loss. Understanding that the protective effects elicited in response to *hsp-3* or *hsp-4* knock-down are orchestrated through the UPR^ER, we next asked if UPR^ER signaling is required for enhanced *F44E5.4* expression. Using quantitative PCR (qPCR), we examined *F44E5.4* transcript levels in Q40::YFP; *fic-1* KO larvae (L4) grown on *hsp-3* RNAi alone or in combination with RNAi against each of the UPR^ER stress sensors. Compared to control RNAi conditions, we again found that *F44E5.4* expression is dramatically upregulated in response to *hsp-3* knock-down (Fig 5E). In contrast, larvae grown on *hsp-3* RNAi combined with RNAi against *ire-1* or *atf-6* showed significant reductions in *F44E5.4* transcript levels (Fig 5E). This effect was most prominent in animals grown on *hsp-3* + *ire-1* RNAi, which showed a significant reduction (approximately 80%) in *F44E5.4* transcripts, while knock-down of *atf-6* resulted in an approximate 50% reduction. Interestingly, combinatorial knock-down with *pek-1* RNAi had no discernable effects on *F44E5.4* expression levels. Correlating this result with our finding that loss of *pek-1* prevents the beneficial effects of *fic-1* KO in development assays, we speculate that this branch of the UPR^ER may regulate expression of other survival factors independent of *F44E5.4*. As a whole, these data mechanistically link UPR^ER signaling through all three branches in the absence of *fic-1* to increased *F44E5.4* expression, blunting misfolded protein toxicity.

Expanding our characterization of this pathway beyond the developmental stages, we found that adult Q40::YFP; *F44E5.4* OE worms showed an increased number of polyQ puncta on par with that observed in Q40::YFP; *fic-1* KO

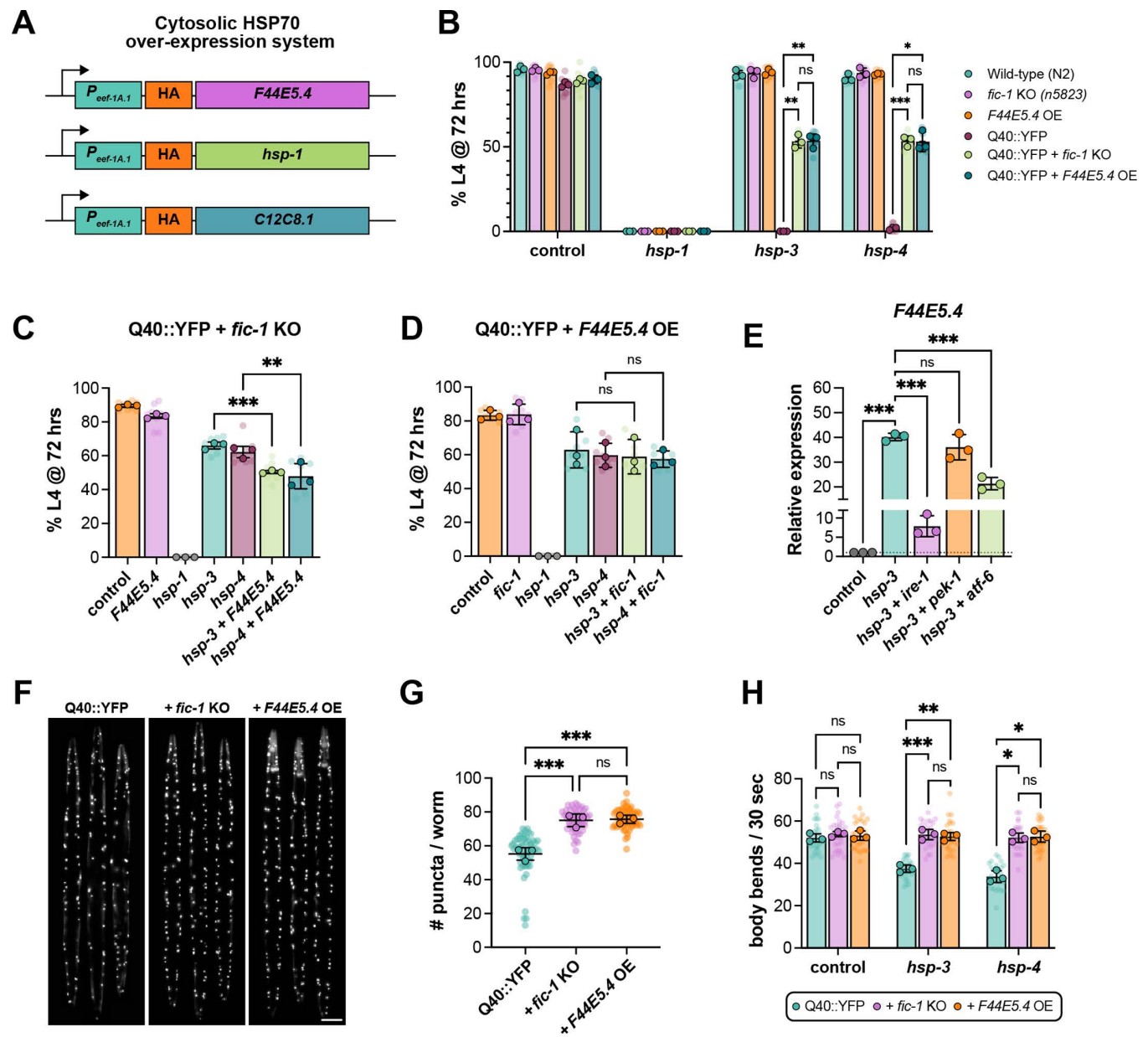

**Fig 5. UPR$^{ER}$ signaling upregulates expression of the cytosolic HSP70 family chaperone, *F44E5.4*, to suppress polyQ toxicity.** (A) Schematic depicting the design of constructs for whole-body over-expression of select HSP70 chaperones. (B) Development assay of the indicated strains comparing survival rates of Q40::YFP+*fic-1* KO and Q40::YFP+*F44E5.4* OE animals. (C-D) Development assays of Q40::YFP+*fic-1* KO animals (C) or Q40::YFP+*F44E5.4* OE animals (D). For all graphs (B-D), X-axes denote the RNAi conditions used. Graphs depict the percentage of animals that have reached the L4 stage of larval development when assessed at 72 hours. (E) Relative mRNA expression levels assessed by qPCR of *F44E5.4* in Q40::YFP+*fic-1* KO L4 larvae fed the indicated RNAis (X-axis), normalized to control. (F) Representative images of day 1 adult Q40::YFP (left), Q40::YFP+*fic-1* KO (middle), and Q40::YFP+*F44E5.4* OE (right) animals. (G) Quantification of the number of polyQ puncta in day 1 adult animals visualized in (E). (H) Quantification of thrashing rates of day 3 adult animals of the indicated genotypes (legend) when treated with the indicated RNAi (X-axis) beginning at day 1 of adulthood (see Fig 2A). For (B-D), translucent data points reflect technical replicates, while opaque data points represent the average of each biological replicate (n = 3). For (G-H), each translucent data point represents one individual worm while opaque data points reflect the average of each biological replicate (n = 3). In (G), at least 50 worms were assessed per genotype. For (H), at least 30 worms per genotype, per condition were scored. Error bars for all plots represent SD. For (B and G), a two-way ANOVA with Tukey's post-hoc multiple comparisons test was performed, and for (C-E, and F) statistical significance was determined using an ordinary one-way ANOVA with Tukey's post-hoc multiple comparisons tests. ***p < 0.001; **p < 0.01; *p < 0.05; ns = not significant.

PLOS Genetics

animals, though no differences in puncta size distribution were noted (Figs 5E–5F, S10C). As an additional measure of adult worm fitness, we assessed thrashing rates. *F44E5.4* OE worms on a wild-type background showed no difference in thrashing rates compared to wild-type (N2) controls (S10D Fig). In day 3 adults on control RNAi, there was no difference in thrashing between Q40::YFP, those lacking *fic-1*, or those over-expressing *F44E5.4*. However, Q40::YFP + *fic-1* KO and Q40::YFP + *F44E5.4* OE animals were both significantly protected from declines in thrashing rates observed in Q40::YFP animals fed *hsp-3* or *hsp-4* RNAi in adulthood (Fig 5G). Taken as a whole, these findings indicate that the absence of *fic-1* under ER stress conditions facilitates UPR^ER signaling to upregulate cytosolic chaperone levels to mitigate polyQ toxicity. Further, the whole-body over-expression of one of these chaperones, *F44E5.4*, is sufficient to avert developmental arrest due to compromised ER homeostasis and protect against fitness declines in adulthood.

## Discussion

The incidence of both pathological and non-pathological protein aggregation increases with organismal age [83]. This is owed to the concomitant declines in functionality of various stress-response pathways, including the UPR^ER and heat-shock response [84–86]. Mounting evidence has implicated the post-translational regulation of HSP70 family chaperones by fic AMPylases in modulating cellular responses to protein aggregates. However, a mechanistic explanation for any these observations has yet to be determined. In this study, we provide evidence that the loss of AMPylation primes the endoplasmic reticulum to initiate protective UPR^ER signaling to combat protein folding stress, resulting in a multifaceted transcriptional response characterized by increased cytosolic chaperoning activity.

Using a well-characterized *C. elegans* model of polyglutamine aggregation, we show that inducing ER dysfunction through RNAi-mediated depletion of either BiP ortholog (*hsp-3* or *hsp-4*) triggers larval arrest in the presence of intact *fic-1* activity. In contrast, animals expressing polyQs in a *fic-1*-null background are viable. We found that, despite the dramatic rescue imparted by the loss of *fic-1*, we did not observe a reduction in polyQ levels or the number of polyQ puncta. One possibility is that the absence of *fic-1* results in changes to polyQ solubility or sequestration, as soluble polyQ intermediates are often considered the primary drivers of cytotoxicity [87]. Further studies into the impact of AMPylation on polyQ protein aggregation, including proteins known to be involved in the interplay between polyQ toxicity and ER stress, such as p97/VCP [88], will be essential for defining the direct impact of AMPylation on these proteotoxic species.

Beyond worm development, we further show that, when this same experimental paradigm is instituted in adult polyQ animals, *fic-1*-null worms are protected from declines in lifespan and fitness observed in the presence of intact *fic-1* signaling. It is interesting to note that the negative effects of *hsp-3* or *hsp-4* depletion, and, relatedly, the rescue imparted by *fic-1* KO, are attenuated in adult animals relative to the effects observed in developing larvae. Previous reports that FIC-1 expression is enriched in *C. elegans* embryos [26] and that constitutive AMPylation is lethal in developing *C. elegans* [42] offer one possible explanation, defining embryonic through early larval development as a critical proteostatic window during which the consequences of deranged AMPylation are especially impactful.

Through bulk RNA-sequencing analysis, we reveal that disruption of ER homeostasis through loss of either *hsp-3* or *hsp-4* in the context of aggregating polyQs has widespread consequences, activating gene programs linked to glutathione metabolism, protein refolding, and UPR^ER signaling. A yet-discussed aspect of these data is that, despite both *hsp-3* and *hsp-4* being considered orthologs of mammalian Grp78/BiP, and previous literature showing compensation by one ortholog upon loss of the other [39,89], knock-down of each elicited distinct transcriptional responses. Specifically, loss of *hsp-3* in Q40::YFP; *fic-1* KO animals resulted in a robust upregulation of numerous molecular chaperones and genes related to glutathione (GSH) synthesis, while *hsp-4* knock-down more strongly induced genes linked to the UPR^ER, ERAD, and N-linked protein glycosylation. Thus, while our prior understanding of these orthologs suggested interchangeable functionality, *hsp-3* and *hsp-4* may play distinct stress-responsive roles to counteract proteostasis stress. The ER matrix is well known as a highly oxidative environment due to its role as a major site for protein folding, resulting in the generation of reactive oxygen species (ROS) in response to ER stress [90]. As such, the *hsp-3* depletion-specific upregulation

of GSH synthesis and chaperone genes may reflect attempts to buffer oxidative stress in the face of increased protein folding demands [91,92]. In contrast, the enrichment of UPR[ER], ER protein processing, and ERAD genes in *hsp-4* deficient Q40::YFP; *fic-1* KO animals formulates a distinct response to ER stress focused on the restoration of ER homeostasis through the clearance of ER client proteins [93]. The role of N-linked protein glycosylation in this paradigm is less clear, though recent studies indicate UPR[ER]-directed changes in N-glycan structural remodeling through XBP-1 may function to communicate ER stress between cells [94]. To this end, the distinct transcriptional changes observed between *hsp-3* and *hsp-4* knock-down conditions may reflect complementary responses working in concert to restore ER homeostasis.

A number of previous studies in the AMPylation field have focused on the implications of FICD-mediated BiP AMPylation for downstream UPR[ER] signaling in mammals. In this study, we show that, in the absence of FIC-1-mediated AMPylation, all three branches of the UPR[ER], controlled by IRE-1, PEK-1, and ATF-6, engage a protective signaling axis to avert developmental arrest in Q40::YFP-expressing *C. elegans* larvae when ER function is compromised. This protective effect of UPR[ER] induction in the absence of AMPylation is mirrored by recent reports that *Ficd*-deficient mice are protected from hypertrophy-induced heart failure and that *Ficd*-deficient cardiomyocytes show enhanced UPR[ER] induction in response to ER stress [95]. Additional studies in *Ficd* KO mouse embryonic fibroblasts (MEFs), *Ficd* KO AR42 cells, and HEK293T cells treated with *Ficd* siRNA similarly showed UPR[ER] induction in response to ER stress, but further indicated that UPR[ER] activation varied in a cell type-specific manner [29,31,96]. While UPR[ER] induction is essential for protection from acute ER stress, ER-mediated apoptotic cell death is a well-known consequence of prolonged UPR[ER] activation [97]. Our finding that *fic-1* deficiency buffers against tunicamycin stress in the absence, but not the presence, of polyQ proteins (S3C Fig) underscores the recently proposed notion that FIC-1/FICD serves to act as a rheostat, fine-tuning the response to ER stress [48]. While we find that the loss of FIC-1-mediated AMPylation is protective in the context of polyQ toxicity, it is not universally beneficial – for example, we previously reported that increased AMPylation is protective in a *C. elegans* model of amyloid-β toxicity [41]. As such, the effects of *fic-1*/*Ficd* deficiency in the face of ER stress are likely dependent on the stressor, duration of stress, and the longevity of the cell types involved. The model we use expresses aggregating polyQs in *C. elegans* body wall muscle cells, which are generally considered to have low turnover. While the UPR[ER] is highly conserved across species, one notable difference in *C. elegans* is the lack of an ortholog to mammalian C/EBP homologous protein (CHOP) [98,99], a transcription factor that is activated downstream of the UPR[ER] to mediate ER stress-induced apoptosis. As such, further studies in physiologically-relevant cell types (e.g., neurons) and higher-order mammalian models of neurodegeneration are needed to determine the extent to which our findings of a UPR[ER]-dependent protective effect from *fic-1* loss are generalizable.

Perhaps our most interesting finding is that over-expression of the HSP70 family chaperone, *F44E5.4*, in the cytosol rescues worms expressing aggregation-prone polyQs from developmental arrest when ER homeostasis is disrupted. From the outset, a major confounding question of ours has been *how* the loss of AMPylation in the ER protects against the toxicity of presumed cytosolic [100] Q40::YFP puncta. In this study, we present a plausible explanation for this phenomenon. Mechanistically, we propose that, in the absence of *fic-1*, the loss of *hsp-3* or *hsp-4* initiates an ER stress response program characterized by activation of all three UPR[ER] stress sensors (IRE-1, PEK-1, and ATF-6), whose downstream effectors translocate to the nucleus to initiate transcription of *F44E5.4* and other stress-responsive genes, dampening polyQ toxicity (Fig 6). Aside from the upregulation of *F44E5.4*, whose activation is coordinated predominantly through IRE-1 and ATF-6, numerous other gene programs likely contribute to the observed rescue phenomenon. Notably, previous studies have linked *Perk*/*pek-1* to the activation of oxidative stress genes [101], including those involved in glutathione biosynthesis [102], presenting a plausible explanation for the upregulation of *gst-* family enzymes we observed upon *hsp-3* knock-down. The activation of ER-associated degradation (ERAD) genes upon *hsp-4* knock-down also represents a significant contributor, though upregulation of these genes is controlled predominantly through the IRE-1/XBP-1 signaling pathway in *C. elegans* [103]. While these are just two examples of additional gene programs involved in the suppression of polyQ toxicity in our model, they underscore the notion that signaling through distinct UPR[ER] sensors can occur synergistically, working in concert to respond to protein folding stress [104].

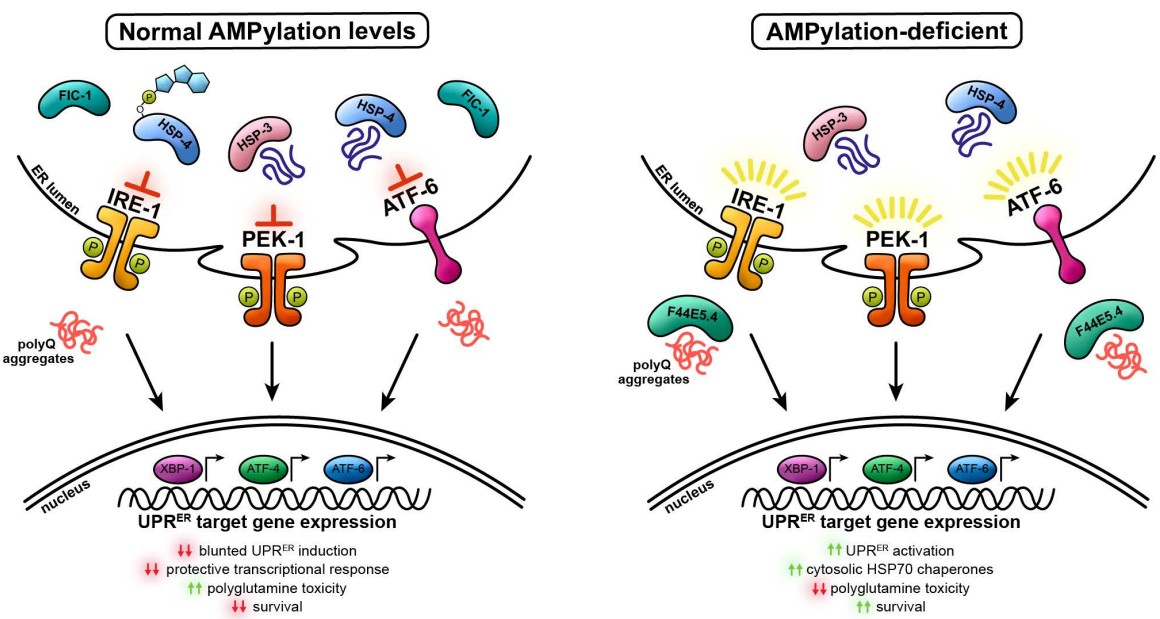

**Fig 6. Summary of proposed mechanism.** In the absence of *fic-1*, disruption of ER homeostasis in the presence of aggregation-prone polyQs activates the unfolded protein response in the endoplasmic reticulum (UPR$^{ER}$). Cooperative signaling through all three branches of the UPR$^{ER}$ initiates a protective transcriptomic response to misfolded proteins, including the upregulation of the cytosolic HSP70 family chaperone, *F44E5.4*, to suppress polyQ toxicity.

Taken collectively, our results thus show a mechanistic link between FIC-1-mediated AMPylation and previously observed effects on neurodegenerative disease-associated polypeptides, underscoring the notion that controlling cellular proteostasis by manipulating levels of AMPylation represents a plausible approach to combating protein misfolding diseases. In the context of polyglutamine expansion diseases, further studies into how AMPylation levels impact toxicity based on cell type and polyglutamine protein context will provide valuable insights as to whether this represents a generalizable, or disease-specific therapeutic approach.

## Limitations of this study

Some *C. elegans* tissues, such as the nervous system and germline, are insensitive to the effects of RNAi-mediated gene knock-down. As such, we are unable to draw conclusions about dependence on tissue-specificity for the pathway outlined in this study. The model we use for polyglutamine toxicity expresses a Q40 repeat peptide. While pathologically-expanded polyQ tracts are a shared hallmark of polyQ expansion diseases, the protein context surrounding this region is increasingly recognized to modulate cellular responses in a disease-specific manner. As such, we cannot exclude, and further hypothesize that the effects of AMPylation on polyQ toxicity are likely influenced by protein context. To validate the beneficial effects of *F44E5.4* upregulation, we generated a whole-body *F44E5.4* over-expression strain. While *F44E5.4* over-expression indeed rescues polyQ toxicity in our model, this whole-body approach prevents us from gathering further mechanistic insights into the cell types or tissues required for this rescue. Lastly, we have previously identified additional targets of FIC-1-mediated AMPylation *in vitro* beyond HSP-3 and HSP-4, including translation elongation factors (EEF-1A.2, EEF-1G, EEF-2), histone H3, and the HSP70 family chaperone HSP-1 (ortholog of human HSC70) [26]. While, unlike HSP-3 and HSP-4, these targets have not yet been shown to be AMPylated *in vivo*, we recognize that *fic-1* deletion may potentially impact additional AMPylation targets that could be involved in proteostasis regulation.

## Supporting information

**S1 Fig. (A-B) Western blots of lysates from L2 larvae fed the indicated RNAis from hatching and probed for Thr-AMP signal (top) and α-tubulin (bottom) as a loading control.** These blots represent additional biological replicates of the data depicted in Fig 1A and are included in the quantification of Thr-AMP signal shown in Fig 1B. (C) Development assay indicating the proportion of wild-type or Q40::YFP animals surviving to the L4 stage of development after 72 hours at 20ºC when fed the indicated RNAis (legend) from hatching. (D-F) Relative mRNA expression levels of the *pos-1* (D), *hsp-3* (E), and *hsp-4* (F) when animals were fed RNAi against the indicated gene from hatching, analyzed by qPCR. Error bars for all plots represent SD. For C, a two-way ANOVA with Tukey's post-hoc multiple comparisons tests was performed to determine statistical significance. For (D-F), statistical significance was calculated using an unpaired T-test. ***$p < 0.001$; ns = not significant.
(TIF)

**S2 Fig. (A-C) Western blots of lysates from L2 larvae fed the indicated RNAis from hatching and probed for Q40::YFP signal using an anti-GFP antibody (top) and α-tubulin (bottom) as a loading control.** These blots represent 3 biological replicates. (D) Quantification of Q40::YFP expression levels expressed as a percent ratio normalized to α-tubulin. Errors bars represent SD. A two-way ANOVA with Tukey's post-hoc multiple comparisons tests was performed to determine statistical significance. ns = not significant.
(TIF)

**S3 Fig. (A-B) Representative depictions of (A) HSP-3 and (B) HSP-4 AlphaFold2-predicted structures highlighting AMPylated residues detected by mass spectrometry.** (C-D) Development assays of the indicated strains (see legend) depicting the percentage of animals reaching the L4 stage of larval development at 72 hours in the presence of the ER stressors (C) tunicamycin or (D) thapsigargin. X-axes denote the control (DMSO) and concentrations tested. For (C-D), translucent data points reflect technical replicates, while opaque data points depict the average of each biological replicate (n = 3). For (C-D), two-way ANOVAs with Tukey's post-hoc multiple comparisons tests were performed to determine statistical significance. *$p < 0.05$; ns = not significant.
(TIF)

**S4 Fig. (A-C) Lifespan curves of polyQ worms in the presence of absence of the *fic-1(n5823)* null allele when fed control (A), *hsp-3* (B), or *hsp-4* (C) RNAi in adulthood.** These graphs represent additional biological replicates of the lifespan curves shown in Fig 2. A Mantel-Cox test was used to determine statistical significance. (D-F) Frequency distribution profiles of polyQ puncta sizes in day 3 adult worms fed control (D), *hsp-3* (E), or *hsp-4* (F) RNAi. Bin size = 50 a.u. (G-I) Frequency distribution profiles of polyQ puncta sizes in day 5 adult worms fed control (G), *hsp-3* (E), or *hsp-4* (F) RNAi. Bin size = 100 a.u. ***$p < 0.001$; ns = not significant.
(TIF)

**S5 Fig. (A-B) Volcano plots showing *hsp-3* and *hsp-4* expression in (A) *hsp-3* or (B) *hsp-4* RNAi-treated Q40::YFP; *fic-1* KO animals relative to control.** Cut-offs (dashed lines): $\log_2 FC > 1.5$, $p < 0.05$. (C) Fragments per kilobase of transcript per million mapped reads (FPKM) values for *hsp-3* and *hsp-4* transcripts in Q40::YFP; *fic-1* KO animals fed the indicated RNAis (X-axis). Each data point reflects one biological replicate (n = 3). (D-E) Relative *hsp-3* (D) and *hsp-4* (E) expression levels in animals fed the indicated RNAis (X-axis). (F-G) Visual depiction of commonly upregulated genes across all genotypes on (F) *hsp-3* or (G) *hsp-4* RNAi vs. control, grouped by gene ontology (GO) biological process terms.
(TIF)

**S6 Fig. (A) Venn diagram of genes downregulated upon *hsp-3* knock-down, with 290 genes specific to Q40::YFP; *fic-1* KO animals.** (B) Over-represented gene ontology (GO) biological process terms downregulated under *hsp-3* knock-down conditions, ordered by fold enrichment. (C) Over-represented GO cellular compartment terms downregulated upon

*hsp-3* knock-down, ordered by fold enrichment. (D) Venn diagram of genes downregulated in response to *hsp-4* knock-down, with 543 genes specific to Q40::YFP; *fic-1* KO animals. (E) Downregulated GO biological process terms under *hsp-4* knock-down conditions, ordered by fold enrichment. (F) Downregulated GO cellular compartment terms in response to *hsp-4* knock-down, ordered by fold enrichment. (G-H) Limited screen of genes downregulated on *hsp-3* (G) and *hsp-4* (H) RNAi performed in Q40::YFP animals. X-axes indicate the RNAi conditions used. Each plot depicts the percentage of animals that have reached the L4 stage of larval development when assessed at 72 hours. For (G-H), each data point reflects one plate, or technical replicate.
(TIF)

**S7 Fig. (A-B) Graphs depicting the log$_2$FC values of select genes upregulated on *hsp-3* (A) and *hsp-4* (B) RNAi.** These plots correspond with the heat maps shown in main-text Fig 3I - 3J. Genes are colored according to functional grouping (from left to right, A: glutathione metabolism, lysosome, protein folding, stress response; B: ERAD pathway, ER stress and protein processing, glycotransferase activity, lysosome, protein folding, stress response, and UPR). (C-D) Limited screen of genes upregulated on *hsp-3* (C) and *hsp-4* (D) RNAi performed in Q40::YFP; *fic-1* KO animals. X-axes depict RNAi conditions used. Each plot shows the percentage of animals that have reached the L4 stage of larval development when assessed at 72 hours. For (G-H), each data point reflects one plate, or technical replicate.
(TIF)

**S8 Fig. (A-B) Heat maps of genes specific to (A) wild-type (N2) or (B) *fic-1* KO *(n5823)* animals on *hsp-3* RNAi vs. control.** (C-D) Heat maps of genes specific to (C) wild-type (N2) or (D) *fic-1* KO *(n5823)* animals on *hsp-4* RNAi vs. control. For all graphs, genes are colored according to log$_2$FC values.
(TIF)

**S9 Fig. (A-E) Relative mRNA expression levels of *ire-1* (A), *pek-1* (B), *atf-6* (C), *atf-4* (D), and *eif-2A* (E) when animals were fed RNAi against the corresponding genes, analyzed by qPCR.** (F) Development assay testing the combinatorial knock-down of *pek-1* and *atf-6*. X-axis indicates RNAi conditions used. Graph depicts the percentage of animals that have reached the L4 stage of larval development when assessed at 72 hours. For (A-E), statistical significance was determined using unpaired T-tests. For (F), statistical significance was assessed using a two-way ANOVA with Tukey's post-hoc multiple comparisons tests. ***p<0.001; **p<0.01; ns=not significant.
(TIF)

**S10 Fig. (A-B) Development assays testing the ability of (A) *hsp-1* OE and (B) *C12C8.1* OE to rescue Q40::YFP larval development under *hsp-3* and *hsp-4* knock-down conditions.** Groupings on X-axes reflect the RNAi condition used. Each graph depicts the percentage of animals that have reached the L4 stage of larval development when assessed at 72 hours. (C) Profile of Q40::YFP puncta size distribution in day 1 adult Q40::YFP, Q40::YFP+*fic-1* KO, and Q40::YFP+*F44E5.4* OE worms. Bin size=5 a.u. (D) Thrashing rates of day 1 adult wild-type (N2) and *F44E5.4* OE animals. (E) Relative *F44E5.4* mRNA expression levels in Q40::YFP+*fic-1* KO animals fed the indicated RNAis (X-axis), normalized to control (*pos-1*). For (A-B), translucent data points reflect technical replicates, while opaque data points depict the average for each biological replicate (n=3). In (C), at least 50 animals were assessed per genotype. For (D), each translucent data point reflects one individual worm, with at least 45 animals per genotype scored. For (E), each data point represents one biological replicate (n=3). For (A-B), two-way ANOVA with Tukey's post-hoc multiple comparisons tests were performed, and in (D) an unpaired T-test was used to assess statistical significance. ***p<0.001; ns=not significant.
(TIF)

**S1 Table. *C. elegans* strains used in this study and their sources.**
(DOCX)

**S2 Table. PCR primers used in this study.**
(DOCX)

**S3 Table. RNAi clones used in this study.**
(DOCX)

**S4 Table. RT-qPCR primers used in this study.**
(DOCX)

**S1 Data. Excel file containing raw data underlying all Main Text figures.**
(XLSX)

**S2 Data. Excel file containing raw data underlying all Supplementary Figures.**
(XLSX)

## Author contributions

**Conceptualization:** Kate M Van Pelt, Matthias Christof Truttmann.

**Data curation:** Kate M Van Pelt.

**Formal analysis:** Kate M Van Pelt.

**Funding acquisition:** Kate M Van Pelt, Matthias Christof Truttmann.

**Investigation:** Kate M Van Pelt.

**Methodology:** Kate M Van Pelt.

**Project administration:** Matthias Christof Truttmann.

**Resources:** Matthias Christof Truttmann.

**Software:** Kate M Van Pelt.

**Supervision:** Matthias Christof Truttmann.

**Validation:** Kate M Van Pelt.

**Visualization:** Kate M Van Pelt.

**Writing – review & editing:** Kate M Van Pelt, Matthias Christof Truttmann.

## Acknowledgments

We thank the members of the Truttmann lab for helpful comments and discussion. We also thank the University of Michigan Advanced Genomics Core for performing the RNA-sequencing and the University of Michigan Proteomics Resource Facility for conducting mass spectrometry of *in vitro* AMPylation reactions.

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
