## [Decision Letter · Decision Letter 0]

PGENETICS-D-24-01488

Loss of FIC-1-mediated AMPylation activates the UPRER and upregulates cytosolic HSP70 chaperones to suppress polyglutamine toxicity

PLOS Genetics

Dear Dr. Truttmann,

Thank you for submitting your manuscript to PLOS Genetics. After careful consideration, we feel that it has merit but does not fully meet PLOS Genetics's publication criteria as it currently stands. Therefore, we invite you to submit a revised version of the manuscript that addresses the points raised during the review process.

Please submit your revised manuscript within 30 days Feb 20 2025 11:59PM. If you will need more time than this to complete your revisions, please reply to this message or contact the journal office at plosgenetics@plos.org. Please include the following items when submitting your revised manuscript:

We look forward to receiving your revised manuscript.

Kind regards,

Javier E. Irazoqui

Academic Editor

PLOS Genetics

Pablo Wappner

Section Editor

PLOS Genetics

Aimée Dudley

Editor-in-Chief

PLOS Genetics

Anne Goriely

Editor-in-Chief

PLOS Genetics

**Additional Editor Comments :**

Please experimentally address the comment regarding RNAi specificity for *hsp-3* and *hsp-4* .

**Journal Requirements:**

At this stage, the following Authors/Authors require contributions: Kate Van Pelt. Please ensure that the full contributions of each author are acknowledged in the "Add/Edit/Remove Authors" section of our submission form.

The list of CRediT author contributions may be found here: https://journals.plos.org/plosgenetics/s/authorship#loc-author-contributions

https://journals.plos.org/plosgenetics/s/submission-guidelines#loc-parts-of-a-submission

5) We notice that your supplementary Figures, and Tables are included in the manuscript file. Please remove them and upload them with the file type 'Supporting Information'. Please ensure that each Supporting Information file has a legend listed in the manuscript after the references list.

Potential Copyright Issues:

i) Figures 2A, 3A, and 6. Please confirm whether you drew the images / clip-art within the figure panels by hand. If you did not draw the images, please provide (a) a link to the source of the images or icons and their license / terms of use; or (b) written permission from the copyright holder to publish the images or icons under our CC BY 4.0 license. Alternatively, you may replace the images with open source alternatives. See these open source resources you may use to replace images / clip-art:

7) In the online submission form, you indicated that "Unprocessed raw datasets (images, excel tables, and prism files) containing and analyzing the data presented in this study are available from the corresponding author upon reasonable request."  All PLOS journals now require all data underlying the findings described in their manuscript to be freely available to other researchers, either

1. In a public repository

2. Within the manuscript itself

3. Uploaded as supplementary information.

8) Please amend your detailed Financial Disclosure statement. This is published with the article. It must therefore be completed in full sentences and contain the exact wording you wish to be published.

9) Please ensure that the funders and grant numbers match between the Financial Disclosure field and the Funding Information tab in your submission form. Note that the funders must be provided in the same order in both places as well. Currently, the order of the funders is different in both places.

Please indicate by return email the full and correct funding information for your study and confirm the order in which funding contributions should appear. Please be sure to indicate whether the funders played any role in the study design, data collection and analysis, decision to publish, or preparation of the manuscript.

**Reviewers' comments:**

Reviewer's Responses to Questions

Reviewer #1: Summary: This manuscript expands on previous findings that fic-1-mediated AMPylation alters polyQ-YFP aggregation dynamics and motility in adult C. elegans. The current work shows that genetic deletion of fic-1 AMPylase rescues developmental arrest, lifespan deficits, and motility impairment in C. elegans with body wall muscle expression of polyQ40 when the worm BiP ortholog hsp-3 or hsp-4 is knocked down, and this rescue depends on intact UPRER sensors ire-1, pek-1, and atf-6. The authors characterize and validate the distinct transcriptomic response of Q40::YFP; fic-1 null worms to hsp-3 or hsp-4 knockdown that appears to facilitate rescue by restoring ER homeostasis. They identify the HSP70 family chaperone F44E5.4 as a gene whose induction depends on intact UPRER sensors and whose overexpression rescues disease polyQ toxicity with hsp-3 or hsp-4 knockdown. This work is impactful in its identification of a suppressor of polyQ toxicity in the context of ER stress that is responsive to AMPylation status, suggesting that modification of AMPylation status or targeting of a single chaperone might be therapeutically useful. However, the paper exhibits limitations in methods and therefore generalizability of these findings, which are made clear in the discussion and limitations sections.

Major concerns:

• This study is heavily dependent upon RNAi methodology and body wall muscle expression of polyQ.

• The specificity of RNAi here is not demonstrated for hsp-3 and hsp-4. These are highly similar orthologs and the claim that RNAi treatment of each is specific for one and not the other of hsp-3/4 must be demonstrated. For example, does hsp-3 RNAi treatment impact hsp-4 transcript levels and visa versa.

• Including validation of at least the main findings using a neuronal polyQ would be more disease relevant and increase the impact of this work. An ideal experiment would be to to recapitulate the RNAi findings from muscle instead in neurons. This could be accomplished using existing hsp-3 and/or hsp-4 loss of function alleles

• If the claim is that fic-1 is beneficial in the context of ER stress, why not induce ER stress genetically using xbp-1s transgenes or pharmacologically as an orthogonal method to RNAi?

• Whole organism overexpression of F44E5.4 precludes the reader from determining whether the phenotypic rescue is cell autonomous or non-cell autonomous since polyQ is being expressed in body wall muscle. Tissue specific rescue would be better.

Minor concerns:

• The language parsing the difference between hsp-3 and hsp-4 as BiP orthologs seems unclear. Please include explanation that hsp-3 as constitutive, hsp-4 as the UPR inducible molecular chaperone.

• “aggregate” is used in the main text but “puncta” is used in methods and figure legends. Would be prudent to clarify the semantic difference (i.e. “aggregate” suggests insoluble, but solubility assay not performed in this paper)

• Lines 345-346 “Combinatorial knock-down of fic-1 + hsp-3 or fic-1 + hsp-4 in wild-type or Q40::YFP animals recapitulated this rescue (Supplementary Fig. S1C), suggesting a cell-autonomous, rather than a neuron-involving cell non-autonomous, mechanism.” Requires clarification or should be preceded by explanation of the limitation of RNAi on neurons. Just because the combinatorial RNAi knockdown recapitulated the rescue does not necessarily mean that the rescue is cell-autonomous.

• Fig.1B-C uses an antibody for Thr AMP but not Ser AMP. Is the reader to assume that Thr AMP and Ser AMP sites are AMPylated at similar rates?

• Does fic-1 deletion rescue polyQ tracts longer than 40 upon hsp-3/4 knockdown? The Q40 is a realtively mild phenotype compared to longer expansions.

• F44E5.4 and C12C8. are both orthologs of HSPA6 and HSPA7 according to WormBase. Could the authors comment on the specificity of F44E5.4 rescue and how is this translatable if the orthologs are the same?

• Please clarify whether there are other AMPylation targets besides BiP, i.e. does fic-1 deletion have other effects? If possible, more clearly state the molecular steps from fic-1 loss to UPR activation via BiP

• Introduction (line 48-51) “One prominent example is the endoplasmic reticulum (ER)-resident HSP70 chaperone, BiP, which can trigger the ER’s unfolded protein response (UPRER) and downstream stress-response signaling through activation of UPRER sensors IRE1 and PERK” should also include ATF6

• Introduction (line 53-54) “However, the consequences of such a strategy, which include the potential to fuel e.g. cancer cells” remove “e.g.”

• Discussion: “multi-facetted” should be “multifaceted” (line 590); instead of “sustained fic-1 activity” would suggest “intact” (line 594); should be “complementary” (line 640)

• A more thorough discussion of similarities and differences between mammalian & worm UPR would be helpful for the discussion. Perhaps, mention lack of CHOP downstream of UPR in worms in discussion (lines 632-651).

Reviewer #2: This study investigates how the AMPylase enzyme FIC-1 interferes with protein homeostasis maintenance during pathologic protein aggregation. Protein AMPylation is an important post-translational modification to regulate the activity of Hsp70 chaperones, in particular for the endoplasmic reticulum Hsp70 BiP, which serves a master regulator of the unfolded protein response (UPR). Many studies, including ones from the senior of author here, have shown how FIC-1 mediated AMPylation is important for modulating the aggregation of neurodegenerative diseases-associated proteins and that FIC-1 regulates UPR signaling. This study now provides further insights how FIC-1 dampens the UPR under proteotoxic stress and is detrimental to organismal development. In a C. elegans model of proteotoxicity from combined polyQ repeat expansion and knockdown of the BiP orthologs hsp3 and 4, the author find that FIC-1 knockout relieves larval arrest. Using transcriptomics and genetic approaches, they establish that the FIC-1 knockout enhances signaling through multiple UPR pathways. Overall, this study is carried out rigorously in most part, the conclusions are supported by the experimental data, and the manuscript is well written. Some limitations are pointed out. The detailed characterization of UPR signaling under hsp3 vs. hsp4 knockdown and discovery of a specific cytosolic chaperone, F44E5.4, which is regulated by the UPR and suppressed by FIC-1, is especially interesting, highlighting adaptability of the UPR and existence of specialized signaling programs. This study should be of interest to a broad readership interested in protein aggregation diseases, proteostasis and stress signaling. The findings have interesting wider implications on whether FIC-1 inhibition can enhance physiology under less construed proteotoxic stress conditions in other organisms.

I just have a few comments that the authors should address prior to publications:

• To cause proteotoxic stress in the Q40:YFP strain, the authors deplete hsp3 or hsp4, which are directly connected to FIC-1 signaling as they are the predominate AMPylation targets. One open question is whether FIC-1 knockout can protect from other proteotoxic stressors through a similar mechanism. I am sympathetic that this is a larger question that may be better suited for a follow up manuscript, but I am curious whether the authors have compared other ER stressors in the Q40:YFP FIC-1 KO vs. Q40:YFP backgrounds.

• In Fig. 1B (and S1A-B), the residual AMPylation in the hsp4 knockout is stronger than in the hsp3 knockout (especially in WT N2). Does this reflect relative expression differences of hsp3 and hsp4, or are the two proteins AMPylated to different levels? This may have important ramifications for which of the BiP orthologs is more sensitive to regulation by FIC-1 and important for activating the specific UPR-ER signaling dependencies identified later. Maybe the authors could use the RNA-Seq data to (crudely) compare the relative hsp3/4 expression to tease these possibilities apart.

• Another questions that remains unaddressed is an explanation for what causes the larval arrest if not the polyQ aggregates themselves? It is somewhat surprising that the levels of polyQ aggregation are not reduced under the rescue in the FIC-1 hsp3/4 double knockout. That being said, it is likely that other proteotoxic species, such as smaller soluble aggregate species, are more detrimental. At minimum, the authors should expand the discussion around this point.

• Does the hsp3 and hsp4 knockdown lead to UPR-ER activation in the absence of polyQ expression? What genes are commonly upregulated in the hsp3 and hsp4 KDs regardless of the strain (center of the Venn diagramm in Fig. 3C & F)?

• Line 488-489: Can the authors clarify what implicated the IRE-1 and PEK-1 branch specifically in the Gene Ontology. This is not clear from the panels.

• For the heatmaps of prioritized genes in Fig 3I-J, the authors should also show the other strain conditions to better highlight that the activation is specific to the Q40:YFP; fic-1 KO condition.

• The authors observed strong induction of both F44E5.4 and F44E5.5, but only tested F44E5.4 for suppression of larval arrest. Did they also investigate F44E5.5. Are these proteins paralogs? How closely related to other Hsp70 chaperones are these?

• What is the knockdown efficiency of F44E5.4 in the combinatorial knockdown experiments? Could incomplete knockdown explain the relatively weak reduction in the L4 rescue?

• I suggest integrating the control qPCR from Fig. S7E to the main text panel Fig. 5E for easier comparison. This will make it clearer that the ire-1 and atf-6 RNAi only incompletely attenuate the F44E5.4 activation.

Reviewer #3: In this study by van Pelt et al., the authors investigate how loss of FIC-1, an ampylase that removes AMP residues from posttranslationaly modified proteins, influences proteotoxicity caused by polyQ40 expression using C. elegans as a disease model. They show that depletion of the ER Hsp70 orthologues in C. elegans, hsp-3 or hsp-4, leads to larval arrest in the PolyQ40 disease model that can be rescued by deletion of fic-1. The authors went to investigate the underlying reason for this, and find that all three UPR ER stress pathways are activated. Interestingly, fic-1 deletion during hsp-3 RNAi leads to an upregulation of the cytosolic Hsp70 isoform F44E5.4, that acts as a critical suppressor of PolyQ toxicity. All in all, the study is well controlled and has a logical flow. It provides important and new insights into the mode and mechanism of why fic-1 deletion has beneficial effects in protein folding diseases which will be a significant addition to our knowledge in the field.

Some additional points need to be clarified:

The study uses the Q40::YFP expressed in bodywall muscle to investigate effects of fic-1 ko. IS this specific for PolyQ or are other disease models similarly impacted? For example amyloid beta (1-42) expressed in the muscle?

One point I found confusing is that the authors say fic-1 k.o. activates all three branches of UPR-ER signaling during development to combat misfolded proteins. But the read-out for statement is the ability of Q40 expressing C. elegans depleted for fic-1 to develop into L4s during RNAi of ire-1, xbp-1, pek-1, atf-6. It is not solely dependent on the upregulation F44E5.4, since pek-1 RNAi in combination with hsp-3 RNAi does not lead to a reduction of F44E5.4. What are other targets controlled by these transcription factors that could potentially contribute to the beneficial effect. This can also be addressed textually.

Minor:

Figure 4. Provide titles for Figures 4A, 4B, 4C to increase clarity.

It is not clear why pos-1 RNAi is used as control (p13, line 352) and Figure S1. This needs clarification in the text.

**Have all data underlying the figures and results presented in the manuscript been provided?**

Reviewer #1: Yes

Reviewer #2: **No: ** The transcriptomics data should be placed in a public repository and the processed dataset included as a SI data table (Excel file).

Reviewer #3: Yes

PLOS authors have the option to publish the peer review history of their article (what does this mean? ). If published, this will include your full peer review and any attached files.

**Do you want your identity to be public for this peer review?** For information about this choice, including consent withdrawal, please see our Privacy Policy .

Reviewer #1: No

Reviewer #2: No

Reviewer #3: No

**Figure resubmission:**
---

## [Decision Letter · Decision Letter 1]

Dear Dr Truttmann,

We are pleased to inform you that your manuscript entitled "Loss of FIC-1-mediated AMPylation activates the UPR^ER^ and upregulates cytosolic HSP70 chaperones to suppress polyglutamine toxicity" has been editorially accepted for publication in PLOS Genetics. Congratulations!

Yours sincerely,

Javier E. Irazoqui

Academic Editor

PLOS Genetics

Pablo Wappner

Section Editor

PLOS Genetics

Aimée Dudley

Editor-in-Chief

PLOS Genetics

Anne Goriely

Editor-in-Chief

PLOS Genetics

Comments from the reviewers (if applicable):

Reviewer's Responses to Questions

**Comments to the Authors:**

Reviewer #1: This work explores how loss of fic-1-mediated AMPylation alleviates proteotoxic stress from polyQ-YFP in adult C. elegans when ER stress is induced by knock down of fic-1 substrates hsp-3 or hsp-4. The authors characterize and validate the distinct transcriptomic response of Q40::YFP; fic-1 null worms to hsp-3 or hsp-4 knockdown that appears to facilitate rescue by restoring ER homeostasis. They identify the HSP70 family chaperone F44E5.4 as a gene whose induction depends on intact UPRER sensors and whose overexpression rescues disease polyQ toxicity with hsp-3 or hsp-4 knockdown. This work is impactful in its identification of a suppressor of polyQ toxicity in the context of ER stress that is responsive to AMPylation status, suggesting that modification of AMPylation status or targeting of a single chaperone might be therapeutically useful. Despite multiple technical challenges, the authors addressed reviewer concerns thoroughly and rigorously with both text changes and supplementary data that present a complete and focused scientific narrative suitable for publication.

Reviewer #2: The authors have done a thorough effort to address all of my concerns, and also the other reviewer's comments. I am supportive of publication.

Reviewer #3: The authors have addressed all my questions and those of other reviewers which significantly improved the manuscript.

**Have all data underlying the figures and results presented in the manuscript been provided?**

Reviewer #1: None

Reviewer #2: Yes

Reviewer #3: Yes

PLOS authors have the option to publish the peer review history of their article (what does this mean? ). If published, this will include your full peer review and any attached files.

**Do you want your identity to be public for this peer review?** For information about this choice, including consent withdrawal, please see our Privacy Policy .

Reviewer #1: No

Reviewer #2: No

Reviewer #3: No

**Data Deposition**

http://datadryad.org/submit?journalID=pgenetics&manu=PGENETICS-D-24-01488R1

**Press Queries**

---

## [Editor Report · Acceptance letter]

PGENETICS-D-24-01488R1

Loss of FIC-1-mediated AMPylation activates the UPR^ER^ and upregulates cytosolic HSP70 chaperones to suppress polyglutamine toxicity

Dear Dr Truttmann,

We are pleased to inform you that your manuscript entitled "Loss of FIC-1-mediated AMPylation activates the UPR^ER^ and upregulates cytosolic HSP70 chaperones to suppress polyglutamine toxicity" has been formally accepted for publication in PLOS Genetics! Your manuscript is now with our production department and you will be notified of the publication date in due course.

With kind regards,

Zsofia Freund

PLOS Genetics

On behalf of:
